# Highly dynamic mechanical transitions in embryonic cell populations during *Drosophila* gastrulation

Juan Manuel Gomez [1] ✉, Carlo Bevilacqua [1], Abhisha Thayambath[2,3], Jean-Karim Heriche [1], Maria Leptin [4,5,6], Julio M. Belmonte[2,3] & Robert Prevedel [1,4,7,8] ✉

During development, three-dimensional morphology arises from the balance of forces acting on cells and tissues, and their material properties. Cellular forces have been investigated, however the characterisation and specification of cell material properties remains poorly understood. Here, we characterise and spatially map in three dimensions the dynamics of the longitudinal modulus at GHz frequencies to characterise the evolving blastoderm material properties during *Drosophila* gastrulation utilising line-scan Brillouin microscopy. We find that blastoderm cells undergo rapid and spatially varying changes in their material properties and that these differ in cells with different fates and behaviours. We identify microtubules as potential mechano-effectors, and develop a physical model to understand the role of localised and dynamic changes in material properties during tissue folding. Our work provides the first spatio-temporal description of evolving material properties during organismal morphogenesis, and highlights the potential of Brillouin microscopy for studying the dynamic changes in cell shape and cell material properties simultaneously.

The regulation of cell shape is critical for the three-dimensional acquisition of morphology across the scales of living matter. Cell shape is determined by a balance of forces, as well as the cell's compliance to deformation, ultimately determined by its material properties[1,2]. The molecular mechanisms underlying the generation and the transmission of forces have been extensively studied[3–7]. However, the dynamics of cell material properties determined during the development of multicellular organisms are less understood. This is largely due to the challenges associated with measuring material properties with high spatial and 3D resolution inside dynamic tissues.

The first major, embryo-scale, morphogenetic event in metazoans is gastrulation, which connects cell fate specification with the coordinated acquisition of particular cellular cell shape behaviours[8]. In *Drosophila*, gastrulation begins with the formation of the ventral furrow (VFF), when ventral cells (mesoderm) accumulate medial-apical actomyosin, causing the apical constriction, shape changes, tissue folding and invagination of the mesoderm in an autonomous manner[3,9,10]. Once internalised, mesoderm cells undergo epithelial-mesenchymal transition (EMT)[11,12] (Supplementary Fig. 1a, b and Supplementary Video 1). The remaining dorso-ventral (DV) cell populations respond differently to the internalisation of the mesoderm. Lateral cells (neuroectoderm) move *in-bloc* towards the ventral midline with minimal changes in their apical cellular geometry (Supplementary Fig. 1a, b and Supplementary Video 1). By contrast, dorsal

[1]Cell Biology and Biophysics Unit, European Molecular Biology Laboratory (EMBL), Heidelberg, Germany. [2]Quantitative and Computational Developmental Biology Cluster, North Carolina State University, Raleigh, NC, USA. [3]Department of Physics, North Carolina State University, Raleigh, NC, USA. [4]Developmental Biology Unit, European Molecular Biology Laboratory (EMBL), Heidelberg, Germany. [5]Director's research, European Molecular Biology Laboratory (EMBL), Heidelberg, Germany. [6]Institute of Genetics, University of Cologne, Cologne, Germany. [7]Epigenetics and Neurobiology Unit, European Molecular Biology Laboratory (EMBL), Rome, Italy. [8]German Center for Lung Research (DZL), Heidelberg, Germany. ✉e-mail: juan.elliff@embl.de; prevedel@embl.de

cells, which will generate the embryonic dorsal ectoderm and the extraembryonic amnioserosa, become squamous[8](Supplementary Fig. 1a, b and Supplementary Video 2). These differential cell shape behaviours during gastrulation suggest variations in the material properties among DV cell populations. Hence, *Drosophila* gastrulation constitutes an excellent model to study the intricate connection between material properties dynamics and the acquisition of specific cell shape behaviours.

A range of experimental methods, often combined with theoretical models or simulations have been used to study the mechanical properties of cells in the early *Drosophila* embryo[8,13–15]. Manipulations with ferrofluids and magnetic beads in cellularising embryos revealed differential mechanical properties of the apical cortex and the cytoplasm[14]. Apical tension fields were probed using laser microdissection along DV cell populations before and upon initiation of gastrulation and integrated into an in silico model that assumed surface tension determine cell shape behaviours, suggesting DV cell populations are mechanically distinct[8]. Recently, a role for viscous shear forces during ventral furrow formation has been proposed in embryos lacking the basal membrane[16]. While these studies have advanced our mechanical understanding of early embryogenesis, they do not provide specific insights into the dynamic or three-dimensional changes in cell material properties that occur during morphogenesis in intact embryos at high temporal and spatial resolution.

Recent progress in Brillouin microscopy has enabled applications to living systems to assess such cell material properties in a volumetric fashion with high spatial and temporal resolution[17]. Brillouin microscopy exploits the inelastic interaction between light and biological matter that causes a shift in photon energy ('Brillouin shift') when photons are scattered by intrinsic collective acoustic vibrations within the sample. The shift in energy of the Brillouin-scattered light is related to the sound velocity of the probed material, which is dependent on its material properties[18,19]. To characterise the material properties, the full elasticity tensor is required, which is inherently dependent on both the probing frequency as well as the wavevector (directionality). Even in an isotropic material, the elasticity tensor comprises multiple independent components, encompassing tensile, shear, and longitudinal moduli, each associated with different stress and strain directions. The longitudinal modulus, defined as the ratio between the uniaxial stress to strain, can be calculated from the Brillouin shift if the refractive index and mass density are known. However, even in the absence of these parameters, the Brillouin spectrum can be used as a proxy of visco-elastic properties[20]. Indeed, in this study we report the Brillouin shift as a proxy to the longitudinal modulus, as commonly done in the field.

Brillouin microscopy differs from more established rheological methods in two main aspects. First, Brillouin microscopy measures the longitudinal modulus whereas established rheological methods such as AFM measure the Young's and/or Shear Moduli[2,18]. Because the longitudinal modulus is typically not accessible by standard techniques (e.g., AFM), it has been less characterised in biological systems. Notably, the Young's, shear and longitudinal moduli describe different responses of materials to particular stresses. They can all be derived from the elasticity tensor[19] but measuring all the independent components is practically not feasible in anisotropic materials such as many biological systems. Second, Brillouin microscopy measures mechanical properties at GHz frequencies, probing material responses on timescales several orders of magnitude smaller (~nanoseconds) than those of cellular-level biological processes (~milliseconds). Despite this, direct comparisons between the two techniques show high empirical correlations between these two regimes[21–24]. Typically, a power law can be observed, both in cells[22] and tissues[21], with an exponent varying in the range of 0.02–0.09. That implies that the relative change in quasi static modulus is often 10 to 50 times larger than the measured relative change in the Brillouin modulus.

Brillouin microscopy offers unique advantages for the study of complex, three-dimensional biological systems, such as non-invasiveness, high spatial resolution, and the ability to capture material dynamics inside living samples. These have led to increasing application across various biological disciplines. These include probing intracellular biomechanics in whole living cells[22,25,26], analysing liquid-to-solid phase transitions in individual subcellular structures[25–27], and biomechanical assessments of tissues in vivo[17,20,28]. While its limitations and assumptions must be carefully considered, Brillouin microscopy provides a powerful platform for exploring the mechanobiology of developmental processes with subcellular resolution, providing insightful information about the material properties of (sub-) as well as cellular compartments. The utility of Brillouin microscopy is particularly evident in scenarios where the target tissue is inaccessible to other contact-based techniques, such as AFM. This is exemplified in studies of *Drosophila* gastrulating embryos, where the blastoderm is encased by the vitelline membrane, rendering direct mechanical access and measurements of the underlying blastoderm unfeasible.

Historically, Brillouin microscopy has been limited to quasi-static applications owing to the weak scattering signal and thus long measurement times, which prevented the study of material properties dynamics at the spatial and temporal scales relevant for morphogenesis (seconds to minutes over hundreds of micrometres). Recently, we introduced a line-scan approach to Brillouin microscopy, termed LSBM, that substantially improved its temporal resolution by two orders of magnitude and decreased substantially phototoxicity, thereby enabling the measurement of dynamics of mechanical properties of comparatively large tissue volumes within biologically-relevant timeframes for fast processes [17].

Here, we used LSBM to characterise and spatially map the dynamics of cell material properties in embryonic cell populations during *Drosophila* gastrulation. We observed distinct Brillouin shift dynamics between mesodermal and ectodermal cells. Specifically, we detected a transient increase in the Brillouin shift within the sub-apical compartment of central mesodermal cells during ventral furrow formation (VFF), coinciding with the reorganisation of sub-apical microtubules. Disrupting microtubules with Colcemid reduced the Brillouin shift during VFF, suggesting that microtubules play a direct or indirect role in determining the material properties of these cells on the timescale of minutes. A physical model of VFF further confirmed the importance of a dynamic sub-apical increase in the longitudinal modulus for fold formation. Our study provides the first comprehensive and spatially resolved description of how embryonic cell material properties evolve during an embryo-scale morphogenetic event and points to microtubules as potential drivers of the rapid and dynamic changes in cell material properties during tissue folding.

## Results

### DV cell populations display differential Brillouin shift dynamics during gastrulation

Cells along the DV axis change their shape in different ways at gastrulation stage (stage 6[29]; Supplementary Fig. 1a–c, Supplementary Videos 1 and 2). We therefore explored if DV cells undergo particular material properties dynamics. We first imaged and analysed the dynamic changes of the longitudinal modulus as indicated by the Brillouin shift within the mesoderm (ventral, Fig. 1a) focusing on a 16-cell wide field of cells along the ventral midline (8 cells on each side of the ventral midline, see Methods; Supplementary Fig. 1a, Supplementary Video 1). We measured the Brillouin shift from the onset of VFF (stage 5b[29]) to the initial phase of epithelial-mesenchymal transition (EMT) (stage 8b[29]).

When gastrulation starts, we detected a transient increase in the Brillouin shift that peaks at the initiation of mesoderm invagination (Fig. 1a, b; $t = 10:35$ min, Supplementary Video 3; RM one-way ANOVA followed by multiple comparisons -FDR corrected-: $p = 0.002$),

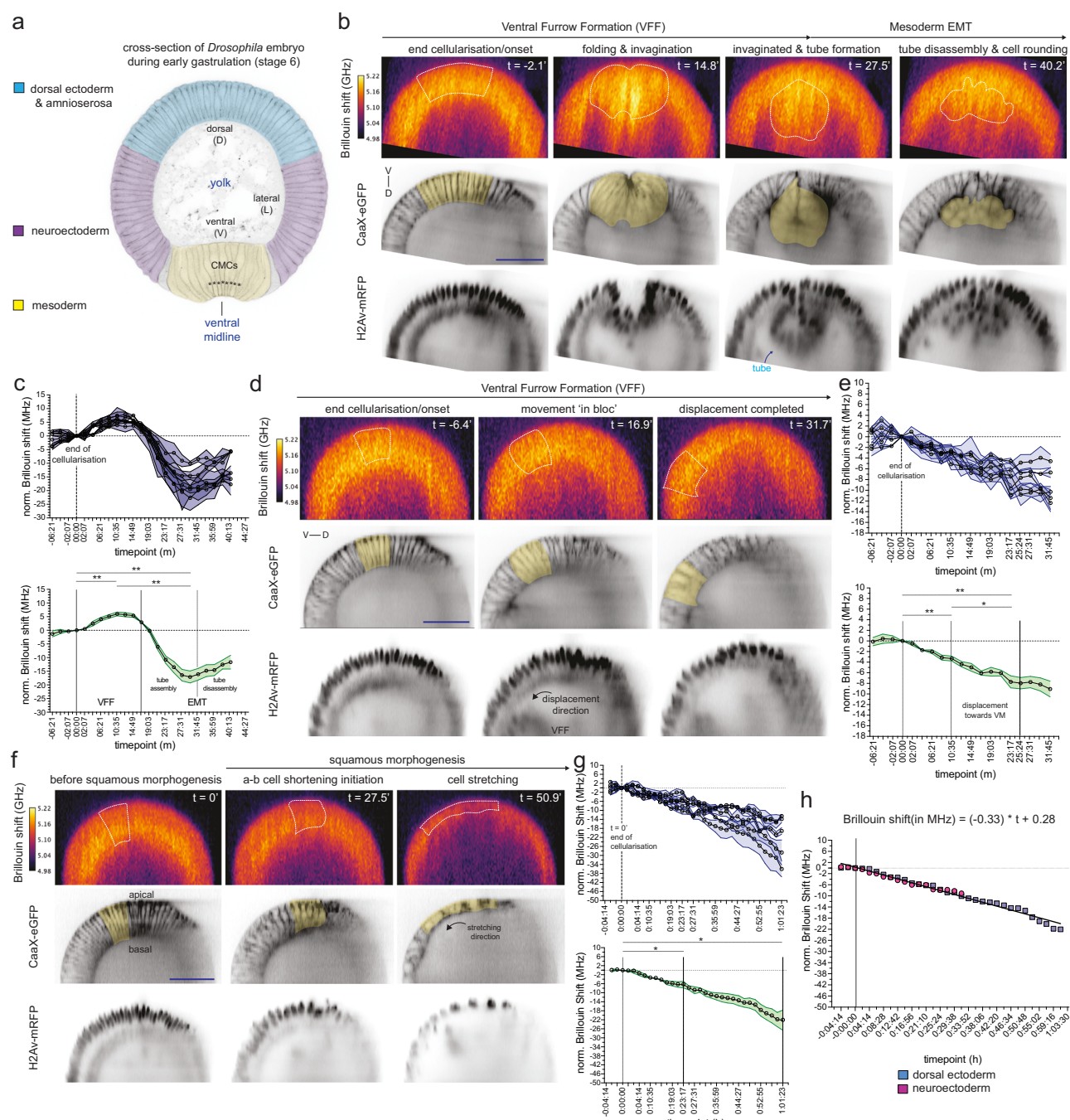

**Fig. 1 | The dynamics of mechanical properties in dorso-ventral cell populations during gastrulation. a** Physical cross-section of a fixed embryo during gastrulation (stage 6: VFF). DV populations depicted: in blue: dorsal ectoderm/amnioserosa; magenta: neuroectoderm; yellow: mesoderm. VM: ventral midline. Asterisks (*) indicate 8 central mesodermal cells. **b** BS maps of the mesoderm at the end of cellularisation (left) and during VFF (centre-left), and EMT (centre-right and right columns). **c** Quantification of the BS within the mesoderm (see Methods). Friedman statistic = 10, $p = 0.0008$. **d** BS maps of the neuroectoderm at the end of cellularisation (left) and during VFF (center and right columns), when it displaces ventrally. **e** Quantification of the BS within the neuroectoderm (see Methods). Friedman statistic = 10, $p = 0.0008$. **f** BS maps of the dorsal ectoderm at the end of cellularisation (left) and during squamous morphogenesis (center and right columns). **g** Quantification of the BS within the dorsal ectoderm (see Methods). $F = 12.05$, $p = 0.0198$. **h** Comparison of BS dynamics between the two subpopulations of the ectoderm: neuroectoderm (lateral; panels **d**, **e**) and dorsal ectoderm (dorsal; panels **f**, **g**). Statistical test: test for combined linear regression model, resulting in a single line: Brillouin Shift (MHz) = $-0.33 \times t + 0.28$; slope comparison: $p = 0.084$; intercept comparison: $p = 0.795$. **h** BS is Brillouin shift. BS imaging (in GHz in top panels in **b**, **d**, **f**) in embryos with fluorescently labelled membranes (CaaX-eGFP: inverted grayscale, middle panels in **b**, **d**, **f**) and nuclei (H2Av-mRFP: inverted grayscale, bottom panels in **b**, **d**, **f**). D: Dorsal; V: Ventral. Quantifications in panels **c**, **e** and **g**: top panels: mean and SEM of the BS (in MHz) in each embryo; bottom panels: mean and SEM of the BS (in MHz) from embryos shown in the top panel ($N = 5$). SEM is standard error of the mean. BS was normalised to the onset of VFF. Yellow area in CaaX-eGFP and white-dashed encircled areas in BS maps indicate quantified areas used in panels (**c**), (**e**) and (**g**). In panels **c**, **e**, **g** the statistical test was: two-tailed RM (Friedman) one-way ANOVA followed by multiple comparisons -FDR corrected-. See main text for the exact $p$-values of multiple comparisons and Methods for details on each analysis; * is $p < 0.05$, ** is $p < 0.01$. Scale bars are 50 µm. Source data are provided as a Source Data file.

consistent with our own previous results[17]. We note that to convert the Brillouin shift to an absolute elastic modulus, the ratio between refractive index (RI) squared and mass density (d) is required. Measuring these parameters independently is difficult in a scattering and thick sample such as the *Drosophila* embryo. However, their ratio only varies a few percent for a wide range of proteins, nucleic acids and sugars[26] and therefore the Brillouin shift can be considered proportional to the longitudinal modulus. The most notable exception from the above assumption are lipid rich compartments[26]. Therefore, one possible cause for the observed increase in Brillouin shift within the mesoderm might be lipid droplets. To rule out this possibility, we studied their distribution in mesodermal cells with a fluorescent marker for lipid droplets (YFP protein-trap in Lsd-2[30,31]). We found lipid droplets along the basal compartment of mesoderm cells during late cellularisation (Supplementary Fig. 1d, Supplementary Video 4) and the initiation of VFF, consistent with earlier observations in EM sections[32] and Raman loss microscopy[33]. However, lipid droplets did not co-localise with the regions of increased Brillouin shift, therefore excluding their presence as the explanation for the increased shift (Supplementary Fig. 1d, Supplementary Video 4). This also further justifies the use of the Brillouin shift as a proxy for the longitudinal modulus in absence of spatial RI and/or d knowledge.

We then analysed the dynamics of material properties in the mesoderm beyond VFF. Once VFF is completed, the invaginated mesoderm transiently arranges in a circular manner that resembles a tube (Supplementary Fig. 1a, Supplementary Video 1) that spans across most of the anterior-posterior axis. When cells initiate the EMT, the tube disassembles, concomitant with the loss of a columnar shape and the acquisition of a roughly spherical geometry (Supplementary Fig. 1a, Supplementary Video 1). After the Brillouin shift peaks in the mesoderm (Fig. 1c; Supplementary Video 3), it decreases concomitant with the completion of mesoderm invagination and the initiation of EMT (Fig. 1b, c; Supplementary Video 3). During this phase cells lose their columnar shape and become round[11] (Fig. 1b; Supplementary Fig. 1a, b). When tube disassembly begins (Supplementary Fig. 1a), and before cells acquire a round shape (t = -31:45 min) mesodermal cells reach a minimum in the Brillouin shift, as indicated by 17 MHz reduction from the initiation of gastrulation (RM one-way ANOVA followed by multiple comparisons -FDR corrected-: $p = 0.0014$) and a 23 MHz reduction in the Brillouin shift (RM one-way ANOVA followed by multiple comparisons -FDR corrected-: $p = 0.0007$) from the initiation of mesoderm invagination, respectively (Fig. 1c).

Next, we analysed the Brillouin shift dynamics within a group of six cells in each of the ectodermal cell populations, the neuroectoderm and dorsal ectoderm. Although these two populations look very similar during the initial phase of VFF (Supplementary Fig. 1a, Supplementary Video 1), the dorsal ectoderm cells, but not the neuroectodermal cells, stretch in the direction of the ventral midline and become squamous throughout stages 7–8 (Supplementary Video 2). From the end of cellularisation, both the neuroectoderm and dorsal ectoderm cells showed a linear reduction in the Brillouin shift (Fig. 1d–g; Supplementary Fig. 2a, b; Supplementary Videos 5 and 6). When neuroectoderm cells have completed their ventral displacement (i.e. end of stage 6), we measured on average a 7.7 MHz reduction in the Brillouin shift (t = 23:17 min; Fig. 1d, e and Supplementary Video 5; RM one-way ANOVA followed by multiple comparisons -FDR corrected-: $p = 0.0027$), which was similar to the reduction in Brillouin shift in dorsal ectoderm cells at the same time (−6.3 MHz, t = 23:17 min; Fig. 1e, g). Squamous morphogenesis continues beyond the period of the ventral displacement of the neuroectoderm. Thus, we followed the Brillouin shift further in time, and measured a reduction of 22 MHz in the dorsal ectoderm cells by the time it had become squamous (t = 01:01:23 h; Fig. 1f, g and Supplementary Video 6; RM one-way ANOVA followed by multiple comparisons, FDR corrected: $p = 0.0156$). Linear regression analyses of neuroectoderm and dorsal ectoderm

cells suggested that the Brillouin shift decreases in a comparable manner in both populations during gastrulation (Supplementary Fig. 2a, b), with the dorsal ectoderm undergoing further reduction in the Brillouin shift -compared to the neuroectoderm- until completion of squamous morphogenesis (Fig. 1e, g). To study whether their Brillouin shift dynamics were truly comparable, we tested if the individual regressions for the neuroectoderm and ectoderm could be described by a combined linear regression model. This showed that the evolution over time of the Brillouin shift in both cell types can be fit by the same line (Brillouin Shift (MHz) = −0.33 × t + 0.28; slope comparison: $p = 0.084$; intercept comparison: $p = 0.795$). These results support a model in which cells along the dorsal-ventral axis exhibit two types of material properties: biphasic Brillouin shift behaviour in the mesoderm and softening at similar rates across ectodermal cells.

## Brillouin shift dynamics in individual cells

Next, we focused on the association between changes in cell shape and the Brillouin shift at the level of individual cells. We again started our analyses with the 16 ventral mesodermal cell rows (Fig. 2a, top panel) in the early phase of VFF. At the end of cellularisation, the Brillouin shift profile was homogeneous (t = −2:07 min, Fig. 2a). When VFF started, the Brillouin shift became heterogenous (t = 2:07 min, Fig. 2a). As VFF progressed, the Brillouin shift decreased slightly in peripheral mesodermal cells (positions |6–8|, Fig. 2a), but increased in cells closer to the ventral midline, i.e., central mesodermal cells, with a peak in the 3 cell rows located closest to the ventral midline (-15 MHz increase, positions |1–3|, Fig. 2a). Central and peripheral mesodermal cells undergo different cell shape changes[6,9]. Central mesodermal cells contract their apical sides, and experienced the largest increase in the Brillouin shift (Fig. 1a, Supplementary Video 1, Fig. 2a -blue arrows-). Peripheral mesodermal cells stretch in the direction of the furrow[34] (Fig. 1a; Supplementary Video 1; Fig. 2a -red arrows-) and showed a reduction in the Brillouin shift. Thus, the Brillouin shift profile in the mesoderm during the early phase of VFF correlates with cell behaviours.

We further explored the subcellular dynamics of the transient increase in the Brillouin shift in central mesodermal cells for which we filtered the pixels in the Brillouin shift maps that had values in the top 4th percentile and analysed their subcellular distribution. When mesoderm folding began, these filtered Brillouin shift pixels were detected between the apical membrane and the apical side of the nucleus, a compartment to which we will refer as 'sub-apical' (Fig. 2b, white dashed line; Supplementary Video 7). When folding progressed to invagination (t = 16.9'; 16:56 min) this entire sub-cellular area accumulated the pixels with highest Brillouin shift values.

Next, we analysed whether the reduction in Brillouin shift during the squamous morphogenesis of the dorsal ectoderm correlates with changes in cellular geometry. When cells transition from a columnar to a squamous shape, their longest (i.e., major) axis is initially the apical-basal axis, whereas the shortest (i.e., minor) is circumferential and along the dorso-ventral direction in which the cell stretches (i.e., 'stretching axis'; Supplementary Fig. 2c). As the cell shortens along the apical-basal axis, it reaches a point where the apical-basal axis length is comparable to the stretching axis (Supplementary Fig. 2c, d). At this point the fitting ellipses become roughly circular (detected using the circularity value; Supplementary Fig. 2d, see Methods), and the major and minor geometric axes of the cell change relative to the directions in the cell. Therefore, we used this point with maximum circularity to operationally divide the process into two phases (Supplementary fig. 2c, d). In the initial phase (phase 1) the columnar cells shortened in the apical-basal axis and expanded in the stretching axis (Fig. 2d, e), generating roughly isotropic cells (Fig. 2d). This was followed by a second phase (phase 2), in which cells stretched along the DV axis and shortened further on the apical-basal axis until cells acquired a squamous shape (Fig. 2d, e).

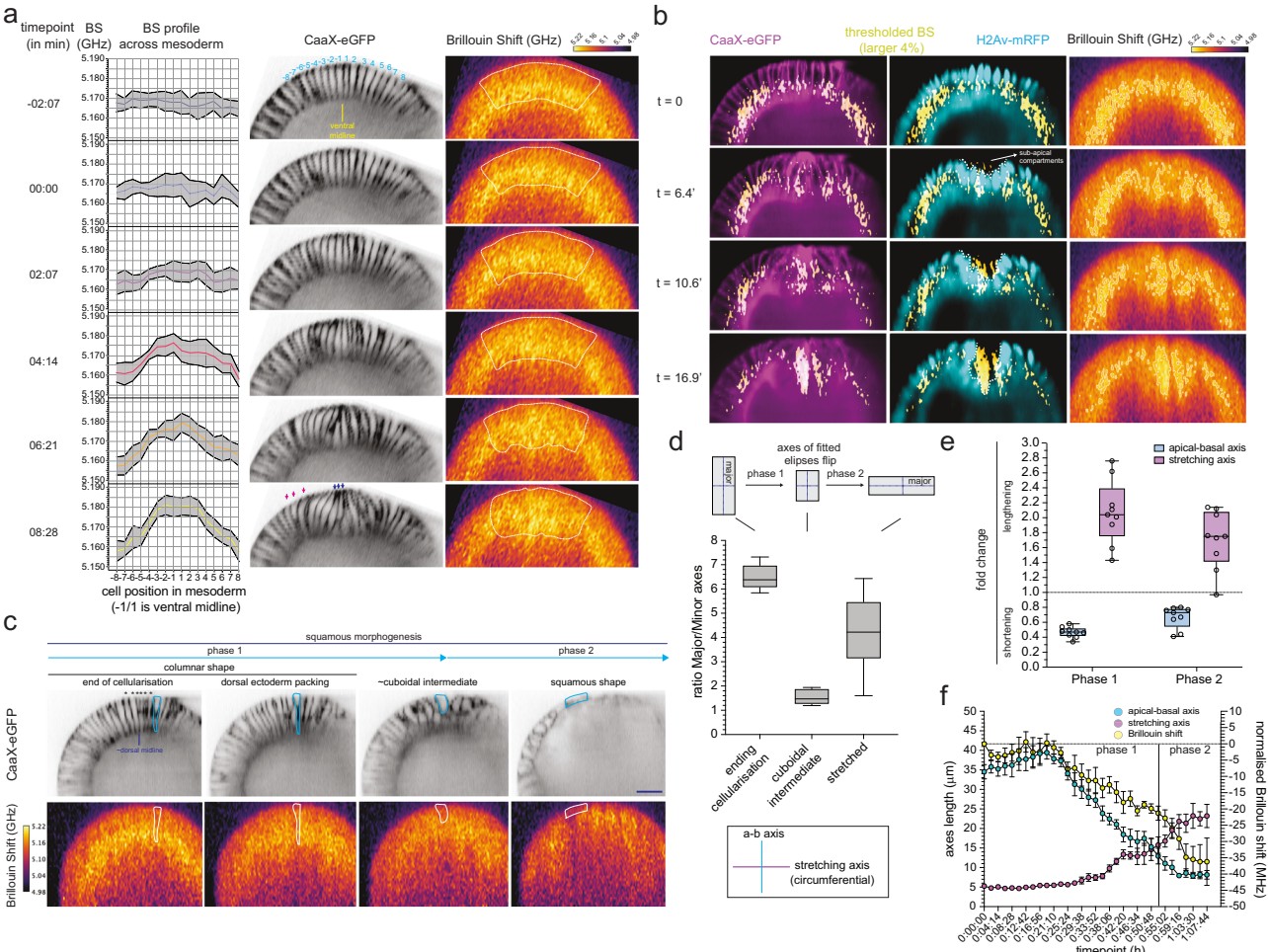

**Fig. 2 | The dynamics of the Brillouin shift at the cellular level. a** Cell-by-cell profile of the BS (in GHz, left column) along 16 cells of the mesoderm (8 cells on each side of ventral midline, center columns) from the onset of gastrulation to ventral fold initiation. Cell position is shown along $x$-axis and within the first timepoint of the CaaX-eGFP column. White-dashed line indicates the quantified 16-cell area. Left column: line is the mean BS shift within each cell, grey area is standard error of the mean (SEM). Three slices in each of four embryos were quantified ($N = 4$). **b** Subcellular localisation of largest 4% of the BS dynamic range during VFF (enclosed with white-dashed line in right column; overlayed in left and center columns). Cell membranes: CaaX-eGFP, in magenta; nuclei: H2Av-mRFP, in cyan. White dashed line in CaaX-eGFP and H2Av-mRFP columns connects the apical side of nuclei across central mesodermal cells. **c** The progression of squamous morphogenesis in dorsal cells: from columnar (left/center-left; blue/white outlines), through a short columnar/cuboidal intermediate (center-right; blue/white outlines) to a squamous shape (right; blue/white outlines). Phase 1 and phase 2 were distinguished by the cuboidal intermediate timepoint (see Methods). Top panel:

membranes labelled with CaaX-eGFP, bottom panel: BS map (in GHz). Asteriks (*) indicate excluded cells from analyses for being considered amnioserosa cells (6 cells). Scale bar is 25 µm. **d** Box-whiskers plot showing major/minor axes length ratio ($N = 9$) of columnar, cuboidal (max. circularity, see Methods) and squamous/ elongated shapes. **e** Box-whiskers plot showing fold changes in apical-basal and stretching axes length from the end of cellularisation until cells reach maximum circularity value (phase 1), and from that timepoint until they become squamous (phase 2). Each dot is a quantified cell ($N = 9$). **f** Time course of the cell-by-cell means of the BS (yellow: normalisation to time = 0': end of cellularisation), and the apical-basal (**a**, **b**, cyan) and stretching (magenta) axes lengths (in µm) during squamous morphogenesis. $N = 7$, (see Methods). Error bars: standard error of the mean (SEM). Box-whiskers plots: line is median, top/bottom edges are 1st and 3rd quartiles and whiskers indicate maximum and minimum values. Source data are provided as a Source Data file. BS is Brillouin shift. For statistical comparisons * is $p < 0.05$, ** is $p < 0.01$.

We detected changes in the Brillouin shift shortly after alterations in the apical-basal axis length (Fig. 2f, $t = 23:17$ min), which persisted as the stretching axis lengthened (Fig. 2f, from $t = 29:38$ min onwards). The Brillouin shift plateaued (Fig. 2f, $t = 59:16$ min) around the time when both the apical-basal (Fig. 2f, $t = 59:16$ min) and stretching axes (Fig. 2f, $t = 57:09$ min) ceased changing. After detrending the time evolution of the Brillouin shift and cell shape parameters (see Methods), we found a correlation of 0.43 between the apical-basal axis and the Brillouin shift, and an anticorrelation of −0.36 between the stretching axis and the Brillouin shift. Overall, our results show that cell shape changes not only coincide with Brillouin shift variations, but also retain a time-independent association, which is slightly stronger for the apical-basal axis.

**The distribution of actomyosin and actin binding proteins does not correlate with the high Brillouin shift measured during VFF**
Changes in elasticity are ultimately mediated by cellular components, with the actomyosin network playing an important role in cell mechanics. Previous studies have shown a link between actin dynamics and the Brillouin shift[22,34,35]. Notably, the Brillouin shift profile (Fig. 2a) closely mirrors the gradient of activated myosin observed within the mesoderm[6]. Mesodermal cells located closer to the ventral midline have higher levels of activated apical actomyosin, and thus, contractility, decreases in a gradient from central to peripheral cells[6]. Therefore, our results point to the cytoskeleton as the cellular component responsible for the measured material properties during VFF and in particular, suggest actin as a possible

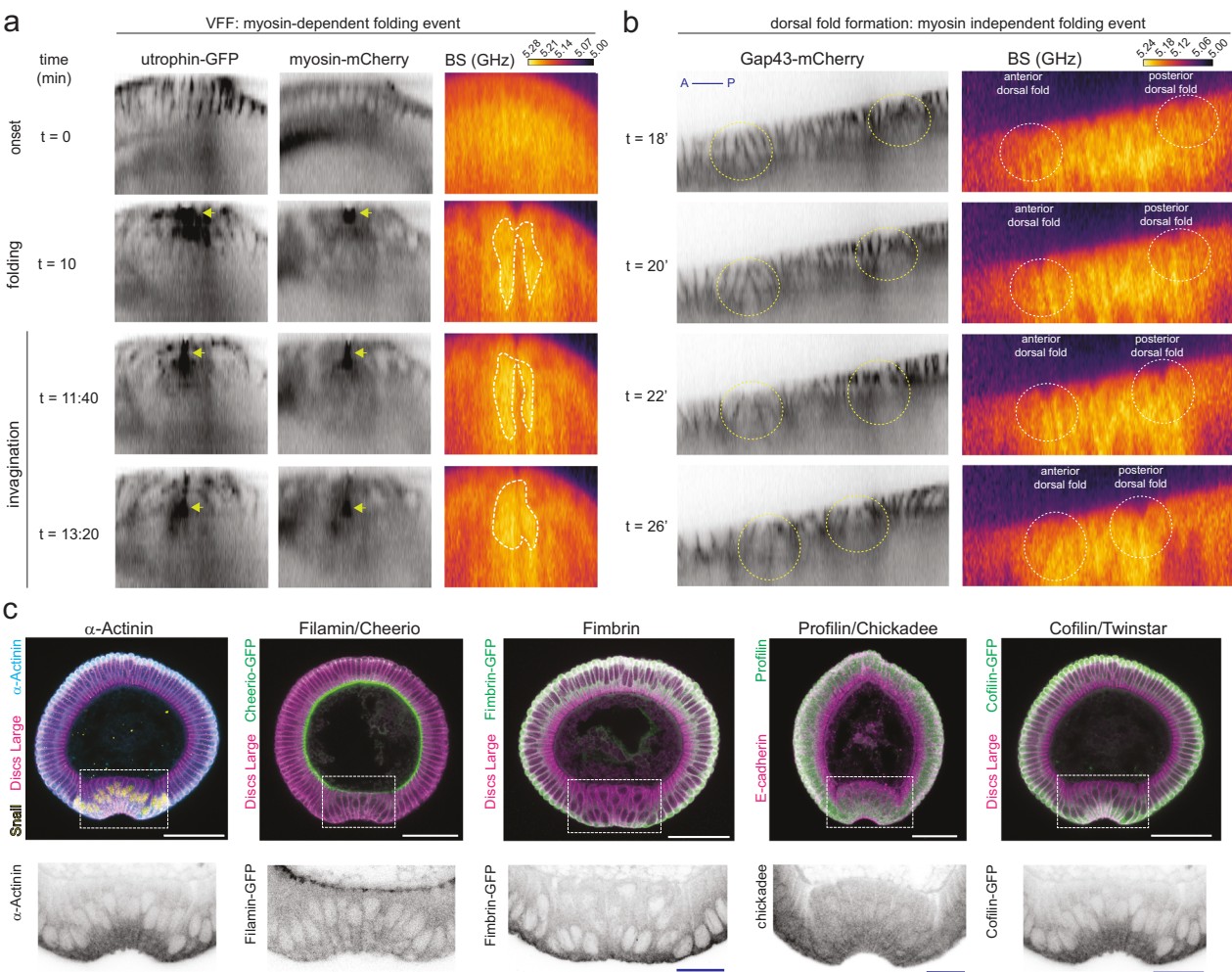

**Fig. 3 | Comparison between actin and actin binding proteins distribution and the transient increase in Brillouin shift. a** Comparison of the localisation of actomyosin and Brillouin shift (right column) during VFF in transgenic embryos with labelled F-actin (utrophin-GFP, left column: inverted grayscale) and myosin light chain (myosin-mCherry, centre column: inverted grayscale). Frames show central mesodermal cells (Fig. 1a, asterisks) at three stages of VFF progression (onset, folding initiation and invagination). Arrows indicate active myosin light chain and F-actin accumulation along the apical side of cells. Encircled area by white-dashed line in BS column indicates the region of the tissue with an increase in the Brillouin shift during folding and invagination. **b** BS shift maps (right panel) from mid-sagittal views of dorsal ectoderm cells (Fig. 1a, blue) during dorsal fold formation (DFF) in transgenic embryos with labelled membranes (gap43-mCherry, left panel: inverted grayscale). Dashed circles (left panel: yellow; right panel: white) indicate tissue areas in which dorsal folds are forming. Time is normalised to the end of cellularisation. Anterior (A) is left, posterior (P) is right. **c** Visualisation of actin binding proteins (top panels: cyan: α-Actinin, green: Filamin/Cheerio, Fimbrin, Profilin/Chickadee and Cofilin/Twinstar; bottom panels: inverted grayscale) in physical cross-sections of fixed embryos undergoing VFF (folding phase). Membranes were stained with antibodies against Discs Large (magenta) except in Profilin/Chickadee stainings, which were co-stained with an antibody against E-Cadherin. Nuclei expressing Snail (yellow) mark mesodermal cells. White dashed squares indicate magnified regions that correspond to the folding mesoderm shown in the insets. Panels show representative images from at least three (3) independent experiments. Scale bars are 50 μm for embryonic D–V cross-section and 20 μm for insets.

contributor to the transient increase in the Brillouin shift within central mesodermal cells.

To explore the association of the actin cytoskeleton with the Brillouin shift dynamics in the mesoderm, we first looked at the colocalisation of actomyosin and the Brillouin shift in embryos carrying fluorescently labelled versions of myosin (sqh-mCherry) and F-actin (UtrophinABD-GFP). When the mesoderm folds ($t=10'$), non-muscle myosin and F-actin accumulate on the apical side of cells[3] (Fig. 3a, arrows; Supplementary Video 8). A small fraction of the high Brillouin shift colocalised with actomyosin during the invagination of the mesoderm ($t=11:40–15:00$ min; Fig. 3a; Supplementary Video 8). However, the transient increase in the Brillouin shift was detected largely basal to the apical cortex, i.e. beyond the area directly at the cortex where actomyosin is concentrated (Fig. 3a, white dashed area; Supplementary Video 8). This finding is consistent with the observation of a transient high Brillouin shift in myosin-independent folding events[36]. During dorsal fold formation (Fig. 3b, Supplementary Video 9), we also observed an increase in the Brillouin shift, thereby further supporting that actomyosin is not the sole cellular component responsible for the transient increase in Brillouin shift.

The dynamic polymerisation and crosslinking of actin filaments with each other regulates actomyosin function and thus, could be the source of the high Brillouin shift observed during VFF. To study this possibility, we analysed the distribution of the bundling actin-actin crosslinkers α-Actinin and Fimbrin, the mesh crosslinker Cheerio/Filamin, and the regulators of F-actin dynamics Profilin/Chickadee and Cofilin/Twinstar within the mesoderm (Fig. 3c, Snail-positive cells) in physical cross-sections of fixed embryos undergoing VFF. These actin binding proteins were found in different subcellular localisations. α-Actinin, Fimbrin, Chickadee/Profilin and Cofilin/Twinstar were present throughout the cytoplasm (Fig. 3c). In contrast, Cheerio/Filamin was enriched in the apical and apical-lateral cortex and at the cellularising

front of blastoderm cells (Fig. 3c). Additionally, Cheerio/Filamin was also differentially distributed along the DV axis (Fig. 3c), suggesting Cheerio/Filamin has different functions in the different cells along the DV axis. None of these actin binding proteins were specifically enriched within the subapical compartment of central mesoderm cells (Fig. 3c). These results suggest that neither crosslinked actin nor actomyosin are directly responsible for the transient increase in the Brillouin shift of central mesoderm cells.

### Investigating the role of microtubules in the dynamics of the Brillouin shift during VFF

The other major system involved in cell mechanics is that of the microtubules. Networks of MT binding proteins and tubulin subunits are regulated through phosphorylation during gastrulation[37], and MTs are involved in controlling nuclear localisation and shape homeostasis in the context of tissue folding [37–39].

We therefore examined the organisation of MTs in physical cross-sections of fixed embryos before and during VFF. Before gastrulation, blastoderm cells have a sub-apical and a basolateral population that differ in their organisations[37,40] (Fig. 4a). At the onset of VFF, the nuclei are located in the wider apical end of the cell (Supplementary Fig. 3a). However, when apical constriction and fold formation starts, the nuclei move basally[37,41,42] (Supplementary Fig. 3a: nuclei labelled with Snail transcription factor, inset: blue dashed-line) together with the centrosomes (labelled with the centriole component Asterless[43]; Fig. 4a, arrows), a movement that has been associated with the basal hydrodynamic flow of the cytoplasm[44,45] and that has been shown to depend on microtubules[37]. The movement of the centrosomes and the nuclei towards the basal side of cells increases the distance between the apical membrane and the nuclei, generating a longer subapical compartment in central mesodermal cells (Fig. 4a, insets, Supplementary Fig. 3a: inset, yellow marked area). During VFF, this enlarged sub-apical compartment became filled with long MTs that aligned with the cellular apical-basal axis in central mesodermal cells (Fig. 4b, top panel), whereas the sub-apical compartment of peripheral mesodermal cells (Fig. 4b, bottom panel) shows microtubules becoming aligned with the apical surface of the cells (Fig. 4b, bottom panel; see also ref. 37).

We also analysed the dynamics of sub-apical MTs in living embryos using spinning-disk confocal microscopy and a transgenic line that labels the MT plus-end tracking protein EB1 (EB1-GFP[46]). To quantify the degree of alignment of MTs with each other, we computed the Microtubule Standard Deviation[47] (MTSD), a metric that measures the variability of the EB1-GFP signal from its average direction. We found that the EB1-GFP signal became increasingly organised within the sub-apical compartment during VFF (Fig. 4c–e, Supplementary Video 11). Most of the reorganisation of the EB1-GFP signal occurred between the onset of VFF ($t = 0'$, MTSD = 77) and the initial phase of fold formation ($t = 6'$, MTSD = 53). During the later folding phase, sub-apical MTs further aligned with each other and the MTSD parameter reached a minimum when folding had progressed to the invagination of the mesoderm ($t = 12'$, MTSD = 48; Supplementary Video 11; Fig. 4d, e, one-way non-parametric ANOVA -Friedman test- followed by multiple comparisons -FDR corrected-: $p = 0.0045$). In summary MTs undergo a fast reorganisation during early VFF, filling the sub-apical compartment of central mesodermal cells with aligned and dynamic MTs. This enhanced alignment of sub-apical MTs relative to each other and to the apical-basal axis in central mesodermal cells is consistent with the reported cell shape-driven mechanism for MT organisation, where MTs become more aligned as the cellular compartment in which they grow becomes increasingly elongated [47].

We reasoned that when sub-apical MTs align parallel to the apical-basal axis, they may increase the Brillouin shift of mesoderm cells along the longitudinal direction, because these are stiff fibres[48]. The alignment of sub-apical microtubules along the cell's apical-basal axis was parallel to the probing direction of the Brillouin

microscope and coincided with the time we measured the peak in Brilloiun shift (Fig. 1b, c; Fig. 4c–e; Supplementary Videos 3 and 11). Thus, these observations are, altogether consistent with a measured increase in the Brillouin shift during mesoderm invagination. Similar results have been previously obtained for ECM fibres ex vivo[49] and in vivo[50]. Next, in order to functionally test the role of MTs in the observed high Brillouin shift during VFF, we depolymerised MTs by delivering Colcemid during mid-to-late cellularisation (see Methods). We focused on the six central mesodermal cell rows with the highest Brillouin shift (Fig. 2a; Fig. 4f, yellow shaded cells). As previously reported, we also observed that Colcemid treatment results in defects in cell shape and nuclear positioning, and ultimately the mesoderm fails to invaginate[37,39] (Fig. 4f, Supplementary Video 10). During the initiation of VFF, the Brillouin shift within the six central mesodermal cell rows of Colcemid-treated embryos increased similarly to untreated embryos (Fig. 4g, Supplementary Fig. 3b). However, Colcemid-treated embryos reached a maximum Brillouin shift that was smaller than in control embryos (median: 16.4 MHz vs. 25.9 MHz, Mann–Whitney test $p = 0.0079$, Fig. 4f, g, Supplementary Video 10). Furthermore, the Brillouin shift in Colcemid-treated embryos stabilised close to its maximum; in control embryos the increase was transient (Supplementary Fig. 3b).

Our findings suggest that the apical constriction and folding of the mesoderm enables an increase in the longitudinal modulus of the central mesoderm through the generation of an elongated sub-apical compartment ranging from the apical cortex to the centrosomes (Fig. 4a, c, -arrows-). This elongated subapical space could enable dynamic MTs to become highly aligned. To test the connection between the formation of the folding event itself and the high Brillouin shift, we analysed *twist* mutant embryos, which are known to impair but not fully abrogate apical constriction, and which engage a few mesodermal cells in fold formation[9] (Supplementary Fig. 3c, Supplementary Video 12). Fold formation is delayed in *twist* mutants ($t = 53.8'$; 53:48 min) compared to control embryos ($t = 20.7'$; 20:42 min) and the transient increase in Brillouin shift is reduced (Supplementary Fig. 3c, Supplementary Video 12). Hence, our results suggest that engaging the whole mesoderm and achieving full contractility are required for the measured increase in the longitudinal modulus as measured by Brillouin microscopy during VFF.

### A physical model of gastrulation shows improved VFF with localized and dynamical changes of cell's longitudinal stiffness

Our observation of transient increases in the Brillouin shift in the context of folding events, including VFF, dorsal fold formation (Fig. 3b), posterior midgut invagination[17] and neurolation[51], suggests that such a transient and spatially restricted increase in longitudinal modulus might be important for the proper progression of folding events. To test this hypothesis, we developed a physical toy model of *Drosophila* VFF within the Cellular Potts framework[52] in which we incorporated known biological qualities such as apical-medial actomyosin contractility, and spatially and temporally varying elastic modulus, as qualitatively informed by our measurements (see Supplementary Note 1). We note that the elastic modulus used in our simulation is quantitatively different from the one measured in Brillouin microscopy, yet aids our understanding about how dynamic changes in cell material properties can be relevant to VFF.

The model simulated the ventral half of the anterior-posterior cross-section of the embryo, encompassing a mesoderm width of 19 cells centred at the VM (Fig. 5a). Here, cells were divided into four compartments: apical, core and basal domains, reflecting the apical–basal polarity of blastoderm cells (Fig. 5b); and a second core domain between the core and basal compartments which increases over time to simulate the observed heightening of the mesoderm cells. A vitelline membrane was added as an outer layer surrounding these cells. Apical constriction was implemented using spring-like forces

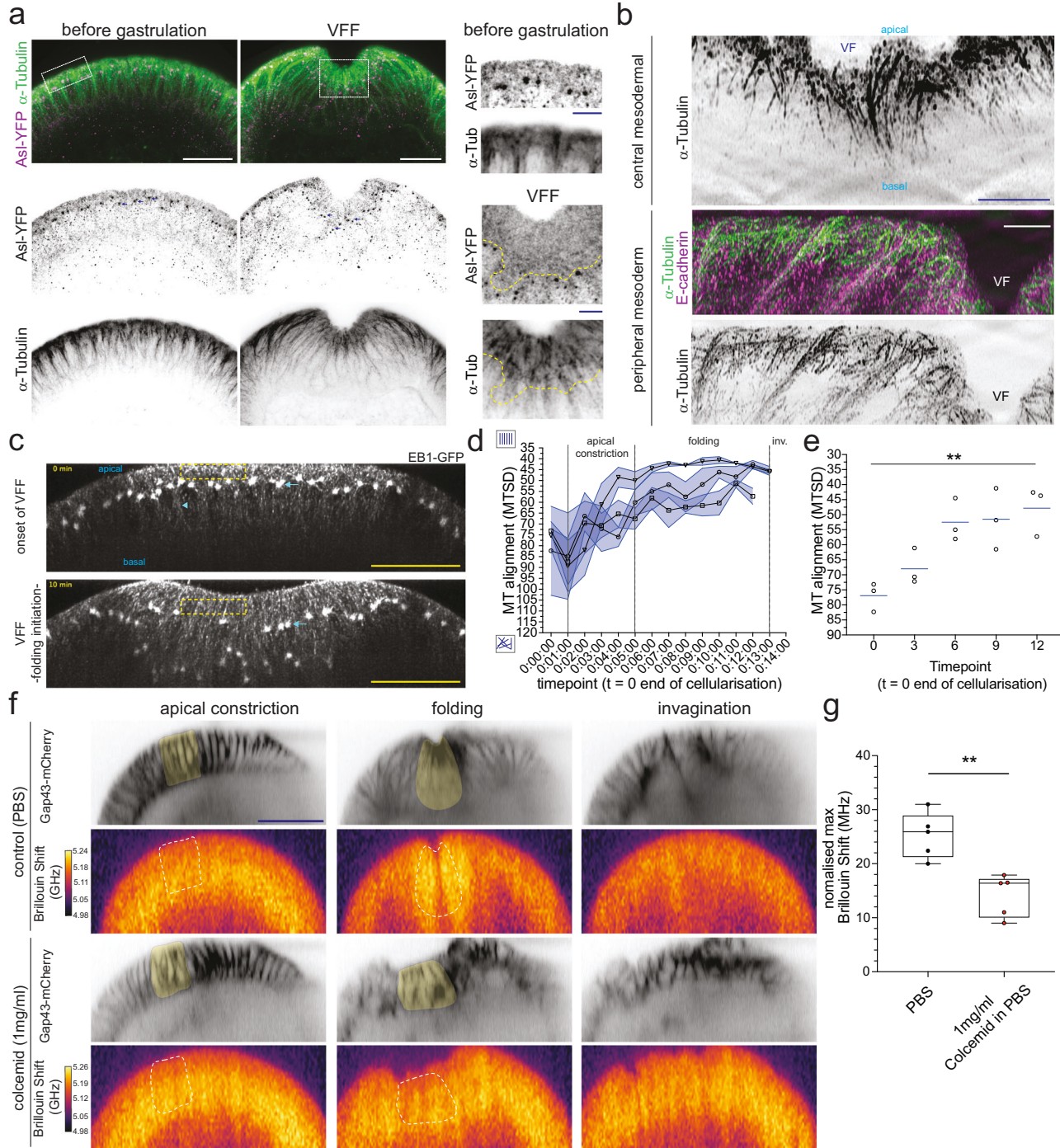

linking the apical sides of the cells (Fig. 5b, first term), where the strength of the contractile force varies over time according to measured myosin levels within the mesoderm[6] (Fig. 5c). Assuming a linear longitudinal stress/strain relation, we use a similar approach to incorporate the longitudinal modulus ($M$) of the cells along their apical–basal axis as the stiffness of 'springs' between the cell compartments (Fig. 5b, second and third terms), where the two $\lambda$ parameters are proportional to the longitudinal modulus ($M$) of the sub-apical and sub-basal regions of the modelled cells (see Supplementary Note 1 in Supplementary Information for a full model description).

We performed stepwise analyses of the effect of the previously identified variations in material properties of cells during VFF: (i) a spatially restricted increase in longitudinal stiffness within each

mesodermal cell, (ii) a stiffness gradient from central mesodermal to peripheral mesodermal cells, and (iii) a dynamic change in stiffness in the order of minutes that recapitulates qualitatively our Brillouin measurements. A comparative analysis of invagination and ingression events revealed that mesoderm invagination during *Drosophila* VFF requires a deeper fold than mesoderm ingression in *Chironomus riparius*[53]. Moreover, both this study and previous work[8,37,39] showed that smaller or shallow mesoderm folds correlate with failed invagination. Consequently, we used fold depth -here furrow depth- (Fig. 5a) as a metric to quantify the extent of fold formation and, therefore, as a proxy for the chance of successful invagination.

When comparing the effect of differential stiffness along the apical–basal axis of the mesodermal cells (Fig. 5d, top) we found that

**Fig. 4 | A role for microtubules in the high Brillouin shift within central mesodermal cells. a** Visualisation of microtubules (α-Tubulin, top: green; bottom: inverted grayscale) and centrosomes (Asterless-YFP, top: magenta; middle: inverted grayscale) in physical cross-sections of fixed embryos at the onset, and during VFF. White squares indicate inset's areas (right column); yellow dashed line: approximate centrosomal position based on Asterless larger puncta. Scale bars are 25 μm; for insets: 10 μm. **b** Super-resolution visualisation of sub-apical microtubules (top: inverted grayscale; middle: green; bottom: inverted grayscale) during VFF. Top: central mesoderm; middle and bottom: peripheral mesoderm. Scale bars are 5 μm. VF ventral furrow. **c** Spinning disk images of the ventral side of living embryos expressing EB1-GFP (grayscale). Top: onset ($t = 0'$); bottom: VFF ($t - 10'$). Dashed-yellow rectangle: sub-apical area used for the analyses of microtubule alignment (MTSD; panel **d**, **e**). Green arrows: centrosomes before (top) and during VFF (bottom). Blue arrow: ventral midline. Blue arrowhead: basal-lateral microtubules. Scale bars are 25 μm. **d** Quantification of MTSD within the sub-apical compartment (yellow dashed rectangle). MTSD was calculated between the onset ($t = 0'$) until fold formation ($t = 13'–14'$). Each line is the MTSD mean per embryo,

with standard error of the mean (SEM, light blue area). **e** Scatter-plot of selected timepoints (panel **d**) of mean MTSD for each embryo (dots) and median across all embryos (blue line). Statistical test: non-parametric one-way ANOVA (Friedman) test, followed by multiple comparisons (FDR corrected). Friedman statistic = 10.67, $p = 0.0040$. $p$ Values of multiple comparisons in Results. **f** BS maps at apical constriction (left), mesoderm folding (centre) and invagination (right) in control (top) and Colcemid-treated embryos (bottom) with labelled membranes (Gap43-mCherry, inverted grayscale). Yellow area (in Gap43-mCherry channel) and white dashed line (in BS channel) indicate quantified areas shown in panel **d**. Scale bar is 50 μm. **g** Box-whiskers plot showing the maximum BS in control (black) and Colcemid-treated (red) embryos. Each dot represents the mean of the 6-cell area in a single slice of an embryo. $N = 5$ independent experiments. Statistical test: two-tailed Mann–Whitney test (non-parametric). Box-whiskers plots: line is median, top/bottom edges are 1st and 3rd quartiles and whiskers indicate maximum and minimum values. Source data are provided as a Source Data file. For statistical comparisons: * is $p < 0.05$, ** is $p < 0.01$.

increasing stiffness within the sub-apical compartment had a substantially larger effect on the furrow depth than increasing sub-basal stiffness (Fig. 5d, bottom). Cells with a very small longitudinal stiffness were too soft and failed to form a furrow. Conversely, cells with increased stiffness within the sub-apical regions always favoured deeper furrow formation for the range of values that we tested, suggesting that cell stiffening is required for tissue folding (see Supplementary Fig. 4a).

To assess the role of the longitudinal stiffness profile along the mesoderm (Fig. 2a), we fixed the sub-basal stiffness and evaluated different sub-apical stiffnesses for the 9 central mesodermal cells and the 5 flanking peripheral cells on each side (Fig. 5e, top). When the central cells had a low stiffness, and thus, were very soft, furrow formation failed regardless of how large the peripheral stiffness values were (Fig. 5e', first 2 panels; Supplementary Videos 13 and 14). Conversely, higher values of central cell stiffness always promote deeper furrows, regardless of the peripheral cell stiffness. We note, however, that for very low values of peripheral cell stiffness furrow formation is prevented by the high deformability of peripheral cells (Fig. 5e', third panel; Supplementary Video 15). Interestingly, deeper furrow formation occurred for combinations of high central cell stiffness with low peripheral cell stiffness, in accordance with our observed changes in Brillouin shift profiles during VFF (Fig. 5e' last panel; Supplementary Video 16). From this we conclude that central mesoderm longitudinal cell stiffness is necessary to translate apical constriction movements into wedge cell shapes that lead to furrow formation, with optimal results obtained with lower values of peripheral mesoderm cell stiffness.

Finally, we demonstrate the importance of dynamic changes in cell material properties by varying the sub-apical longitudinal stiffness of the mesodermal cells over time (Fig. 5f, top panel). Starting from a constant value for all cells, we varied the sub-apical stiffness over time to resemble the dynamics seen in the experiments, with peripheral cells stiffening over time, followed by central cells softening over time (Fig. 2a; Supplementary Fig. 4c). Decreasing peripheral cells' stiffness hinders furrow formation in most regimes and increasing central cells' stiffness promotes furrow formation (6% to 43% increase in furrow depth, depending on the starting sub-apical stiffness value). The best results were obtained with peripheral cells decreasing their stiffness up to 33% and the central cells increasing their stiffness above 200% from the starting value (see highlighted area in Fig. 5f and corresponding snapshot in Fig. 5a'). Most strikingly, in most cases, dynamically changing cell material properties over time produces deeper furrows than starting all the cells with a pre-established stiffness profile, suggesting that dynamical changes of the sub-apical longitudinal modulus of mesodermal cells favour furrow formation (Supplementary Fig. 4d–f).

## Discussion

Using line-scan Brillouin microscopy, we identified rapid and spatially distinct changes in the Brillouin shift across different cell populations in the *Drosophila* embryo during gastrulation. These changes indicate divergent longitudinal moduli between the mesoderm and the ectoderm, that coincide with changes in shape along the longitudinal direction (i.e., apical-basal), suggesting that the longitudinal modulus is relevant for the characterisation of material properties in this morphogenetic event. Notably, mesodermal cells exhibited a biphasic dynamic of their material properties, characterised by an initial increase in the Brillouin shift, followed by a reduction during epithelial-mesenchymal transition (EMT). By combining genetics, live-cell and super-resolution imaging, as well as chemical perturbations, our results suggest that microtubules act as direct or indirect regulators of the transient increase in the Brillouin shift, and consequently, the longitudinal modulus, within the sub-apical compartment of central mesodermal cells. Furthermore, our physical model of ventral furrow formation supports the requirement for localised and dynamic changes in the longitudinal modulus to drive furrow formation and facilitate tissue folding.

The timescales of the cellular and subcellular processes that regulate morphogenetic events, in particular those that are driven by cell shape changes, range from milliseconds to minutes. However, Brillouin microscopy measures the longitudinal modulus in the GHz regime, and thus, measures molecular mechanics that occur on nanosecond time-scales[18,54]. While Brillouin spectroscopy was shown to be able to reveal both the elastic and viscous properties of biopolymers that are central to the structure and function of biological tissues[22,23,55], this difference in timescale still raises the question whether Brillouin microscopy is suitable for studying morphogenesis[20,23]. Previous work has compared Brillouin microscopy with lower frequency techniques such as Atomic Force Microscopy (AFM), a technology that measures the Young's modulus on time-scales more closely related to those of morphogenesis. While earlier work found positive correlations between the Brillouin frequency shift and the Young's modulus in cells[22] and tissues[20,21], recent work highlights the need of combined use of both methods for a more comprehensive mechanical characterisation of biological matter [56].

The entire (volumetric) mix of material content of a cell or tissue will determine its mechanical property; therefore, a cell's solid-to-fluid ratio can have a pronounced effect that can be measured by Brillouin microscopy. For instance, the transient increase in the Brillouin shift within the sub-apical compartment of central mesodermal cells could, in principle, result from a local rise in protein abundance, altering the refractive index or density. However, such a transient (≤15 min) and localised increase in protein concentration within the sub-apical compartment is highly unlikely and has not been reported at the

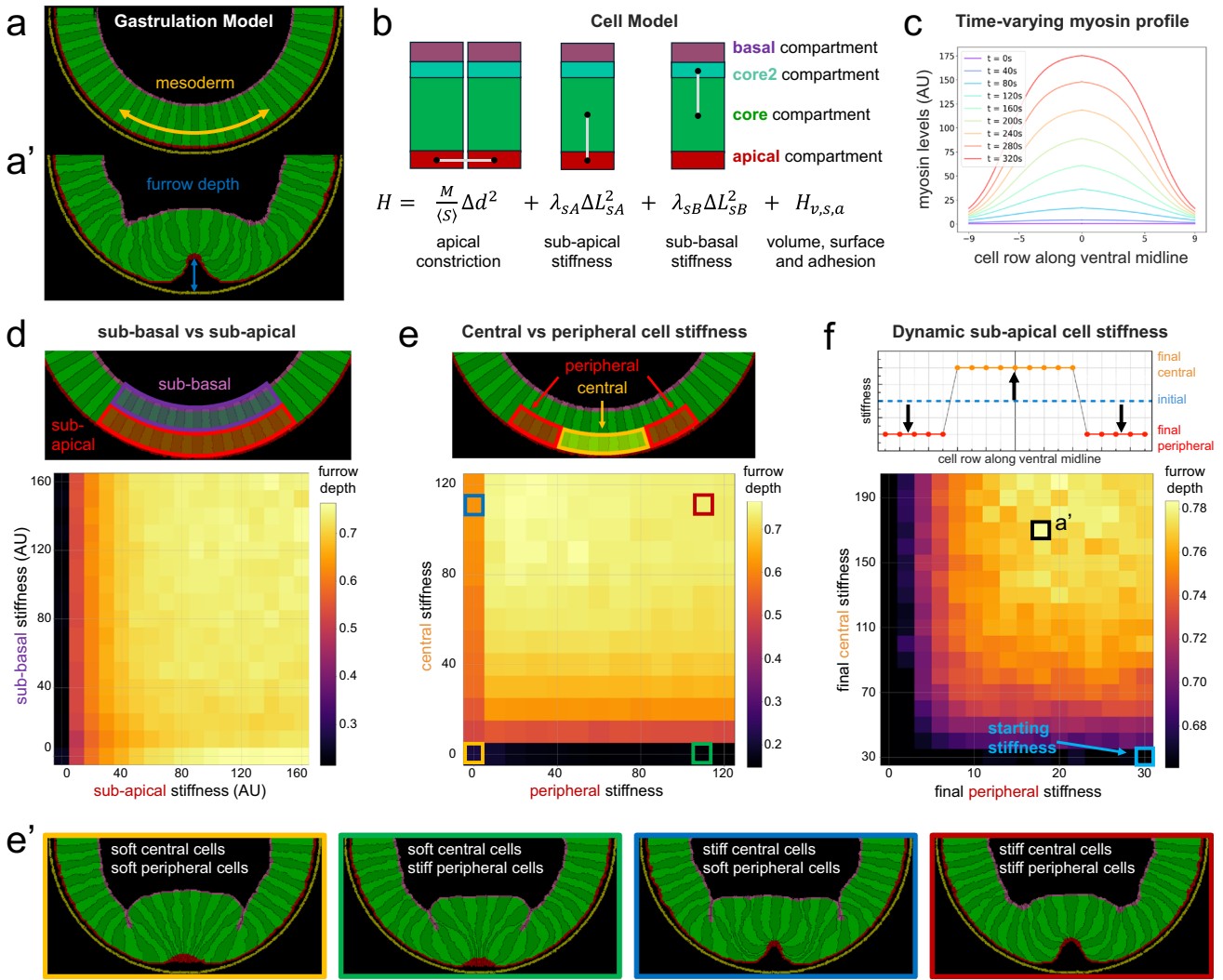

**Fig. 5 | A physical model of *Drosophila* ventral furrow formation. a** 2D model of the ventral half of the *Drosophila* embryo with 19 mesodermal cells (yellow arrow range) and 28 neuroectodermal cells. **a'** Furrow depth metric, measured as the normalised distance between the apical-most coordinate of the central meso-dermal cell and the lowest point of the mesoderm from the initial condition in units of initial apical-basal cell height. Snapshot taken from a simulation corresponding to highlighted values in panel (**f**). **b** Each cell contains 4 compartments: apical, basal and two core compartments, where core2 increases over time in order to repro-duce the observed cell heightening during VFF. Hookean links between compart-ments' centre of mass are used to model apical constriction, and sub-apical and sub-basal longitudinal stiffness. Those constraints are part of an effective energy equation (*H*) that guides the model evolution and also includes adhesion, surface and volume terms. **c** Smoothed profiles of experimentally measured[6] time varying myosin concentrations used to impose apical constriction on mesodermal cells. **d** Furrow depth as a function of sub-apical and sub-basal longitudinal stiffness of mesodermal cells. Furrow depth correlates with higher sub-apical stiffness, but not

with sub-basal stiffness. **e** Furrow depth as a function of sub-apical longitudinal stiffness of central and peripheral cells. When either cell population is too soft, furrows fail to form. Higher values of central cell stiffness promote furrow forma-tion, with optimal results when peripheral cell stiffness is lower (but not zero). Sub-basal stiffness was the same for both cell populations. **e'** Representative simulation snapshots of 4 cases from panel (**e**) (indicated by corresponding outline colour). **f** Furrow depth as a function of increasing sub-apical central cell longitudinal stiffness, and decreasing sub-apical peripheral cell stiffness. Grid coordinate indi-cates final values of central and peripheral cell stiffness. Colour bar cropped at the furrow depth value of simulations where stiffness of both cell populations remains constant at 30. Sub-basal stiffness was kept constant and the same for both cell populations. Highlighted value at (peripheral: 18, central: 170) indicates simulation sets with deepest furrow. A snapshot of a simulation with these values is shown in panel (**a'**). **d**–**f** Each grid point value shows the average of 30 simulations. Source data is provided as a Source Data file.

gastrulation stage. Moreover, this effect would need to be confined not only to the sub-apical compartment but also specifically to central mesodermal cells, where we observed the Brillouin shift increase during VFF (Fig. 2a, b). At this stage, the embryo is undergoing maternal-to-zygotic transition, and the proteomes of dorsal-ventral cell populations display very small differences in abundance, including cytoskeletal components and its associated proteins[37]. Thus, sig-nificant protein abundance differences are unlikely within the meso-derm. Additionally, central mesodermal cell volume remains constant during VFF[44], reinforcing that the observed Brillouin shift dynamics

likely reflect changes in cellular material properties that do not have their origin in altered solid-to-fluid ratios.

Our mechanical characterisation of cells along the dorsal-ventral axis fits well with measurements of mechanical properties using AFM in cells undergoing similar biological processes. The softening of cells undergoing EMT (Fig. 1b, c) is consistent with the reduction in the elastic modulus that was observed in various models of cancer inva-siveness and EMT[57–59]. The stiffening of contractile mesodermal cells during VFF is consistent with an increase in the elastic modulus in contractile cells, such as cardiac and skeletal muscle cells[57,60], although

the mechanisms determining these mechanics may be different in each tissue. Furthermore, an in silico model suggested cells in the follicular epithelium that are committed to become squamous increase their compliance to deformation[1], which is also consistent with the measured softening of dorsal ectoderm cells. In general, our results should therefore be consistent with mechanical transitions that were measured with a force-exerting and lower frequency methodology such as AFM.

The most complete mechanical characterisation of dorsal-ventral cells that preceded our work was conducted by Rauzi et al.[8]. In this study, the authors characterised the mechanics of dorso-ventral cells at gastrulation stage using laser microdissection along the apical cell surface. Assuming cellular mechanics arise from differential cell surface tensions, the authors conclude that ectodermal and mesodermal cells soften whereas neuroectodermal cells stiffen during gastrulation[8]. How can this be integrated with our results? Laser microdissection measures the cortical tensions (i.e., a force) in a direction tangential to the apical surface of the epithelium, potentially averaging contributions from a larger field of cells. In contrast, Brillouin microscopy measures material properties orthogonally to the apical surface and across the entire cell volume with a sub-cellular resolution determined by the axial size of the point-spread function, which is in the order of 3 µm. Therefore, we are unable to specifically resolve the mechanics of the cellular membrane and its associated actin meshwork, in contrast to laser microdissection analyses. An additional difference between both studies is the time-resolution: whereas laser microdissection can be performed only once and in few isolated places, using Brillouin microscopy we capture and spatially map the dynamics of whole cell material properties in the same specimen at regular timepoints. We therefore consider it is indeed plausible that the apical surface of neuroectoderm cells is stiffer compared to that of ventral and dorsal cells while they soften along the longitudinal direction (apical-basal axis). This highlight -again- the importance of combining methodologies to characterise mechanical transitions at the tissue and organismal level.

We describe different mechanical dynamics for the mesoderm and the ectoderm (Figs. 1 and 2). Although both ectodermal cell sub-populations, the neuroectoderm and the dorsal ectoderm, differ in biological qualities that could affect cell mechanics, such as the accumulation of actomyosin and the localisation of adherens junctions[5,47] and Cheerio/Filamin (Fig. 3c, top), we found similar mechanical dynamics among cells in the neuroectoderm and the dorsal ectoderm. Therefore, from a mechanical standpoint, we have identified two distinct domains along the DV axis: the mesoderm and the whole ectoderm. A genetic mechanism that could assign different mechanical properties to mesoderm and the whole ectoderm must rely on the mesodermal determinant Snail[61]. However, it is unclear what are the possible mechanoeffectors that may be regulated by Snail in the mesoderm. A recent proteomic and phosphoproteomic study of DV cell populations at gastrulation stage has found networks of proteins and phosphoproteins whose abundance is increased either in the whole ectoderm or in the mesoderm[37]. Future work will investigate the role of these regulated proteins and phosphoproteins in determining the mechanical dynamics of the ectoderm.

In pre-cellularising Drosophila embryos, the elastic modulus of the cytoplasm is determined by F-actin[14]. However, the region where we detect an increase in the Brillouin shift within mesoderm cells does not fully colocalise with F-actin nor a panel of actin-binding proteins (Fig. 3a, c). Instead, our results point to a mechanical role of microtubules during VFF, as Colcemid treatment impairs the Brillouin shift increase within central mesodermal cells. However, the effect of microtubules on the Brillouin shift during VFF could be manifold: First, they could have a direct effect, acting as structural mechanoeffectors that transmit forces[39] or resist compression[62,63]. Alternatively, their effect could be indirect, by perturbing a downstream process facilitated by microtubules. Microtubule depolymerisation impairs cell shape and nuclear position across dorso-ventral cell populations, and specifically, prevents mesoderm invagination[37,39]. Therefore, it remains a possibility that perturbing mesoderm fold morphology and invagination in Colcemid-treated embryos (Fig. 4f) causes the reduced Brillouin shift, rather than the structural function of microtubules themselves. However, microtubules have properties that support their function as mechano-effectors: high persistent length[48], dynamic instability[64] and differential stiffness[65,66]. Notably, dynamic microtubules are stiffer than stable ones, gaining flexibility through acetylation[65]. Microtubules in the lateral-basal compartment of mesodermal cells are acetylated, whereas sub-apical microtubules are not acetylated and remain tyrosinated[37]. Thus, the progressive alignment of dynamic and stiff microtubules is consistent with our measurements of an increase in the Brillouin shift within the sub-apical compartment of central mesodermal cells. We speculate that the alignment of sub-apical microtubules enables central mesodermal cells to resist compression in the longitudinal direction, thereby facilitating the progression of the folding event.

Our physical model of ventral furrow formation supports a role for the longitudinal modulus, which is derived from our Brillouin measurements. Additionally, the physical model shows that dynamic and localised changes in cellular material properties, which so far have been assumed to be constant during a morphological process, promote the progress of folding events, even though the model only utilises our measurements in a qualitative way and does not physically simulate the high-frequency aspects of the Brillouin scattering process itself. The model predicts that only changes in sub-apical regions of the epithelial cells affect furrow formation (Fig. 5d). This fits well with our observations that Brillouin shifts and MT reorganisation are restricted to that compartment, as well as with previous measurements in cellularising Drosophila embryos that showed a heterogeneous Young's modulus along the pre-blastoderm cytoplasm[14]. While experimental evidence and previous models ascribe a driving function to apical constriction in furrow formation, our model showed that this process is contingent on the longitudinal stiffness of central cells (Fig. 5e'). A stiffer mesoderm along the apical-basal axis correlates with deeper furrows, but the model predicts better outcomes if cells are initially softer and stiffen over time (Supplementary Fig. 4d), again a phenomenon we also see in the Brillouin measurements. This observation of spatially restricted cell stiffening therefore fits with a model in which cell-shape-driven morphogenetic events require cells to behave not as a uniform and constant mechanical unit.

We propose a model in which the observed changes in the material properties of the mesoderm along the longitudinal direction arise from the reorganisation of microtubules. In epithelial tissues, microtubule organisation is dictated by the degree of cell elongation, with microtubules becoming increasingly aligned as cells elongate[47]. We speculate that a similar mechanism regulates the alignment of sub-apical microtubules as the distance between the nuclei and the apical surface increases during VFF, leading to the elongation of the sub-apical compartment (Supplementary Fig. 3a). This process could facilitate microtubule alignment along the apical-basal axis, filling the sub-apical compartment with stiff polymers[48]. Because Brillouin microscopy measures mechanical properties in a directional manner, parallel to the apical-basal axis in this case, the increasingly aligned microtubules within the sub-apical compartment (Fig. 4c–e) are strong candidates for dynamically stiffening mesodermal cells, thereby promoting fold formation. In summary, our results suggest that microtubule reorganisation is ultimately driven by apical constriction and the elongation of central mesodermal cells, a cost-effective mechanism that has also been observed in the Drosophila epidermis later in development [47].

Altogether, our results show that cells can adjust their material properties within minutes to optimally drive morphogenesis, and

highlights the potential of Brillouin microscopy as a powerful tool for investigating the mechanical aspects of cell shape behaviors and their relationship to tissue dynamics in vivo. Future work will determine the exact role of microtubules and their binding proteins in determining the evolving mechanical properties during tissue folding.

## Methods

### *Drosophila* genetics, embryo collections and live imaging mounting

**Drosophila genetics.** To visualise cell membranes and nuclei (Fig. 1b–h, Fig. 2a–c), we used transgenic lines expressing CaaX-eGFP[37] and H2Av-mRFP (both stocks provided by Yu-Chiun Wang). To study the distribution of lipid droplets, we used a YFP protein-trap for the gene *lsd-2* (provided by Yu-Chiun Wang, source is *Drosophila* Kyoto Stock Center/DGGR-Kyoto: 115301)[67].

To analyse the colocalisation between actomyosin and the Brillouin shift during ventral furrow formation (Fig. 3a), we combined the sqh-sqh::mCherry transgene (Bloomington stock N°: 59024) to visualise non-muscle myosin with a utrophinABD-GFP transgene[8].

To study the distribution of actin binding proteins we used the following transgenic lines: GT{cher[mGFP6-1]}/TM6[68,69] a GFP knock-in that specifically labels the long isoform of Cheerio (provided by Sven Huelsmann), PBac{768.FSVS-0}Fim[CPTI003498] (DGGR-Kyoto: N°: 115478) a protein-trap for Fimbrin, PBac{754.P.FSVS-0} tsr[CPTI002237] (DGGR-Kyoto: N°: 115280) a protein-trap for Cofilin/Twinstar. For immunostainings of actin binding proteins, we used either *w*[1118] (Bloomington stock: 5905), or Gap43-mCherry[70] (provided by Stefano De Renzis), a transgenic line labelling cell membrane.

To study the centrosomes position (Fig. 4a) we used a transgenic line (genomic construct) that labels the centriolar protein Asterless (pUbi-YFPAsl[FL])[43].

For microtubule perturbation experiments (Fig. 4f, g), EB1-GFP visualisation (Fig. 4c–e) and analyses of the Brillouin shift during dorsal fold formation (Fig. 3b), we used a transgenic line that combined Gap43-mCherry with EB1-GFP[46] (provided by Nick Brown) to label the plus ends of the microtubules.

To study the connection between the folding and invagination of the mesoderm with the measurement of an increase in the Brillouin shift we used the *twist*[171] mutant allele *twi*[1]/Cyo,hb-GFPnls; 3X-CaaX-mScarlet[34]/TM6ref (provided by Yu-Chiun Wang).

**Embryo collections.** For all live imaging experiments (using either line-scan Brillouin, spinning disk or 2-photon microscopes), we synchronised egg collections for 1 h. The collected embryos were allowed to develop a further hour at 25 °C. The embryos on the agar plates were covered with halocarbon oil (27S, Sigma) to hand-select embryos at the cellularisation stage (stages 5a, b[29]) using a Zeiss binocular. Finally, embryos were dechorionated (standard bleach, 50% in water) for 90 s, washed in PBS 1× and mounted for microscopy.

For immunohistochemistry experiments, we synchronised egg collections for 2 h, and the collected embryos were allowed to develop further for 2 h, dechorionated and transferred to a glass vial for immediate fixation (see immunohistochemistry).

**Live imaging.** For Brillouin microscopy experiments, dechorionated embryos were oriented to target the corresponding cell population (dorsal, lateral or ventral) and glued (heptane glue) on the surface of a custom-built tool made of a plastic material: polyether ether ketone. The temperature of the imaging chamber was 21 °C ± 1.5 °C.

For vertical mounting in combination with 2-photon microscopy, dechorionated embryos were glued (heptane glue) on their posterior end to 35 mm glass-bottom petri dishes (MatTek, 10 mm microwell, N° 1.5 coverglass), with their anterior side pointing up, and embryos were oriented vertically to acquire a dorso-ventral cross-sectional view[37]. Finally, the embryos were embedded in 1% low-point melting agarose

(Sigma-Aldrich, A9414) in PBS added at ~32 °C, and the agarose was allowed to cool to RT (22 °C).

For Spinning Disk microscopy, dechorionated embryos were oriented with their ventral side to be in contact with the glass bottom of the petri dish and 6 ml of PBS were added.

### Immunohistochemistry

**Fixation procedures.** To detect and visualise microtubules and actin binding proteins while preserving the overall cellular structures, a formaldehyde-methanol sequential fixation was performed. Dechorionated embryos were fixed in 10% formaldehyde (methanol free, 18814 Polysciences Inc.) in PBS:Heptane (1:1) for 20 min at room temperature (RT). To visualise microtubules, we used ice-cold methanol:heptane and embryos were stored for 24 h at −20 °C and rehydrated before use[47]. To visualise actin binding proteins, we devitellinised for 45 s in 1:1 methanol:heptane (RT methanol).

**Antibody staining procedures.** Rehydrated embryos were blocked for 2 h in 2% BSA (B9000, NEB) in PBS with 0.3% Triton X-100 (T9284, Sigma). Primary antibody incubations were done overnight at 4 °C. Primary antibodies used were: mouse anti α-tubulin 1:1000 (T6199, clone 6-11B-1, Sigma), rabbit anti Snail[17] 1:500, rabbit anti GFP 1:1000 (ab290, Abcam), mouse anti FITC-GFP 1:250 (ab6662, Abcam), anti E-cadherin 1:50 (clone DCAD2, DSHB), anti Profilin/Chic (clone Chi-1J 1:50, DSHB), anti discs-large 1:50 (clone 4F3, DSHB) and anti α-Actinin 1:100 (ab50599, Abcam). Incubations with secondary antibodies were performed for 2 h at RT. Alexa Fluor 488, 568, 594 and 647-coupled secondary antibodies were used at 1:600 (488: goat anti mouse IgG ab150117, Abcam- and goat anti rabbit IgG ab150081, Abcam; 568: goat anti rat IgG JacksonImmunoResearch, 115-295-166-; 594: goat anti mouse IgG ab150120, Abcam; 647: Goat anti rat IgG 112-605-167, JacksonImmunoResearch and goat anti mouse IgG JacksonImmunoResearch, 115-295-166).

**Preparation of physical cross-sections.** Physical cross-sections of embryos were made as previously described[37]. Embryos that had been immunostained were embedded in Fluoromount G (SouthernBiotech 0100-01) and visually inspected to identify embryos at stage 6 (gastrulation[29]). The selected embryos were sectioned manually with a 27 G injection needle at approximately 50% embryo length and slices were mounted on the sectioned side.

### Light sheet, confocal, spinning disk and 3D-SIM super-resolution image acquisition

The Brillouin microscope is coupled to a light sheet microscope for mapping Brillouin shift spectra to cellular and extra-cellular components (see below 'Brillouin microscopy' section). Brillouin shift and corresponding fluorescent images shown in Fig. 1b are median-projections of three-consecutive slices, images in Fig. 1d, f, Fig. 3a, b and Fig. 4f are median-projections of two consecutive slices.

Live imaging of CaaX-eGFP; H2Av-mRFP embryos (Supplementary Fig. 1a and Supplementary Video 1) was performed with a Zeiss LSM780 NLO 2-photon microscope with a Plan-Apochromat 63× objective (NA 1.4, oil, DIC M27) at a room temperature of 22 °C. We acquired 4 slices (z-slice size = 1, 1 μm) at approximately 180 μm from the posterior end of the embryo every 2 min. Supplementary Video 1 was generated by max-projecting 3 consecutive slices.

Images in Figs. 3c and 4a and Supplementary Fig. 3a were acquired with a Leica SP8 microscope equipped with a supercontinuum laser. Gated detection on HyD detectors was used for each channel using a Plan-Apochromat 63× oil (NA 1.4) objective at 22 °C, with a z-slice size of 0.3 μm. Acquired volumes used in Fig. 3c and Supplementary Fig. 3a were max-projected (along the z-axis) for a range of approximately 1.5 μm (5 slices); volumes used in Fig. 4a were max-projected (along z-axis) for a range of approximately 3 μm (10 slices).

Live imaging of EB1-GFP embryos (Fig. 4c–e and Supplementary Video 11) was performed with an Olympus IXplore SpinSR, with a pinhole size of 50 μm, using the C488-561 dual filter and equipped with 2× ORCA-FLASH 4.0v3 cameras. For acquisition we used a 100× UPLSAPO 100 XS objective (NA 1,35) with silicon immersion. Acquired volumes were YZ resliced and max-projected over 20 consecutive slices. Imaging was performed at a room temperature of 22 °C. The acquired raw volumes for each embryo were deconvolved using Huygens Professional 23.1.

Images in Fig. 4b were acquired using a super resolution Deltavision OMX 3D-SIM (3D-SIM) V3 BLAZE from Applied Precision/GE Healthcare, and correspond to max-projections of 30 consecutive slices. Deltavision OMX 3D-SIM System V3 BLAZE is equipped with 3 sCMOS cameras, 405, 488 and 592.5 nm diode laser illumination, an Olympus Plan Apo 60× 1.42 numerical aperture (NA) oil objective, and standard excitation and emission filter sets. Imaging of each channel was done sequentially using three angles and five phase shifts of the illumination pattern. The refractive index of the immersion oil (Cargille) was 1.516. Acquired volumes were max-projected (along the $z$ axis) for a range of 9 μm (30 slices). Imaging was performed at a room temperature of 20 °C.

## Brillouin microscopy

The microscope combines a self-built fluorescence light-sheet and a confocal line Brillouin modality as described in detail in Bevilacqua et al.[17]. In brief, two water immersion objectives (Nikon MRD07420, 40× 0.8NA) are mounted at 90°. For the fluorescence modality, one of the two objectives generates a static light sheet of approx. 200 μm width and 4 μm thickness and the fluorescence light is detected by the other objective and imaged on a sCMOS (Zyla, Andor). For the Brillouin modality a confocal line is generated by a cylindrical lens in combination with the objective, while the Brillouin back-scattered light is collected by the same objective and sent to a custom-built Brillouin spectrometer. Note that the lenses $CL_2$ and $CL_3$ in Bevilacqua et al.[17], are substituted by a single spherical lens of 150 mm, giving a pixel size along the line of 1.1 μm instead of 0.7 μm. The total laser power on the sample was kept below ~22 mW. The embryos were glued on a 3D-printed tool with a cut at 45°, to facilitate the orientation of the embryos with respect to the detection objective. The tool was connected to a 3-axes translational stage and immersed directly in PBS. For the permeabilized embryos a 12.7 μm thick FEP foil was placed between the embryos and the detection objective to reduce the volume of PBS in contact with the embryos (see below: Microtubule perturbation using Colcemid). In all panels in which the Brillouin shift is shown as normalised (or norm.), the Brillouin shift raw value at the end of cellularisation/onset of gastrulation was substracted from the Brillouin shift raw value at each timepoint.

## Microtubule perturbation using Colcemid

Synchronised egg collections of 45′ were allowed to develop for 2 h at 25 °C. Embryos were dechorionated in 50% bleach in water. We removed the wax layer that covers the vitelline membrane[72] by fully immersing embryos in a permeabilising solution, i.e. EPS[72], 1:10 in PBS for 3 min. The EPS solution consists of 90% D-Lemonene (Sigma-Aldrich, 183164), 5% cocamide DEA (Ninol 11CM, Stepan Chemical, Northfield, Illinois) and 5% ethoxylated alcohol (Bio-Soft 1–7, Stepan Chemical, Northfield, Illinois). Next, embryos were promptly washed first in Triton 0.3% in PBS and subsequently in PBS only. Permeabilised embryos were visually inspected using a Zeiss stereoscope, and based on morphological criteria[29] embryos in stage 5a[29] were selected and mounted. Finally, mounted embryos were placed within the Brillouin microscope incubation chamber filled with either PBS or a solution of 1 mg/ml Colcemid in PBS (Demecolcine, Sigma-Aldrich, D7385). Embryos that were in mid to late cellularization (membrane growth was visualised using the membrane marker Gap43m-Cherry) at the

moment of initiation of the incubation with Colcemid were imaged throughout the gastrulation stage. Five independent experiments, each experiment with the acquisition of one embryo, were performed each for control (permeabilized + PBS) and treated embryos (permeabilised + 1 mg/ml Colcemid in PBS). Imaging quantitation and statistical analyses were performed as described below.

## Image quantification

Image analyses were performed using ImageJ/Fiji version 2.14.0/1.54f and specifically for Fig. 4d, e, we also used Matlab R2019b (see below: Quantification of the alignment of microtubules with each other).

Masks were hand-generated in ImageJ/Fiji to filter the Brillouin signal of target tissue areas. For all quantitation with the exception of EB1-GFP analyses (Fig. 4c–e, see below: Quantification of the alignment of microtubules with each other), masks were generated based on the signal of a membrane marker (CaaX-eGFP or Gap43-mCherry) using the 'polygon selections' tool. We extracted shape parameters of the mask itself and values of the Brillouin shift signal within the region of interest (ROI) that was enclosed by the mask with the ImageJ/Fiji 'measure' tool. The shape parameters of the masks that were estimated were the major and short axis lengths and the circularity values. From the filtered Brillouin shift values, we estimated the median, mean, standard deviation, maximum and minimum Brillouin shift values. Mask generation and the above quantifications were performed on single slices from YZ resliced raw volumes.

**Mask generation and measurements within region of interest.** For the results shown in Fig. 1b, c, a mask to segment 16 cells with its centre in the presumptive ventral midline was generated manually for each timepoint from the onset of gastrulation to mesoderm invagination using the CaaX-eGFP signal for the identification of cell boundaries. Once the mesoderm was invaginated, the CaaX-eGFP signal corresponding to cells deep inside the embryo scattered, difficulting the delimitation between the interior-most cells and the yolk for the generation of masks. To circumvent this problem, we generated a mask by thresholding the CaaX-eGFP signal at the time-point the folding is initiated. The threshold was set to filter out any signal that does not correspond to membranes in any timepoint subsequent to the invagination of the mesoderm. Once the mask was generated, it was overlaid onto the CaaX-eGFP signal and used as a guide to delineate a boundary between the invaginated mesoderm and the yolk for the generation of masks corresponding to the mesoderm cells undergoing EMT.

For the results shown in Fig. 1d, e, a mask consisting of 6 cells positioned at a distance of 11–13 cells dorsally to the boundary of the ventral fold (counted using the nuclear signal from the H2Av-mRFP marker) was generated for each time-point from the onset of gastrulation to the point where the ventral displacement of neuroectoderm cells is complete.

For the results shown in Fig. 1f, g we generated masks for 6 cells that are positioned in the presumptive dorsal ectoderm for each timepoint from the end of cellularisation (when the membrane signal stops growing towards the yolk, Supplementary Video 2) until squamous morphogenesis generates stretched cells. We excluded from our analyses the presumptive amnioserosa, which we considered to be 6 cells wide around the dorsal-midline (3 cells on each side of the presumptive dorsal-midline; Fig. 2c).

For the results shown in Fig. 2a, a mask was generated for each cell of the mesoderm (along a 16-cell field with its centre in the presumptive ventral midline), from the onset of gastrulation until the initiation of the folding event.

For the results shown in Fig. 2c–g, a mask was generated for a single cell for each time-point from the end of cellularisation until the squamous shape had been acquired. We wanted to analyse the correlation between the Brillouin shift and the length of the apical-basal axis

during squamous morphogenesis, using the major and minor axes measurement derived from fitting ellipses with ImageJ/Fiji together with the circularity measurement (Supplementary Fig. 2c; d: 1–3). At the onset of squamous morphogenesis, the apical-basal axis aligns with the major axis of the fitted ellipse, while the perpendicular 'stretching axis' corresponds to the minor axis. As cells adopt an approximately isotropic shape, these axes switch: the major axis becomes the 'stretching axis,' and the minor axis becomes the apical-basal axis. We identified this transition by measuring the 'circularity' of the fitted ellipses and divided the process into two phases: Phase 1, from the initiation of squamous morphogenesis until cells reach isotropy, and Phase 2, from isotropy to the completion of morphogenesis (Supplementary Fig. 2d: step 3). The time point at which the fitted ellipses reach maximum circularity marks the transition from Phase 1 to Phase 2 (Supplementary Fig. 2d: steps 4–5). To track the apical-basal axis in each cell, we combined major axis measurements from Phase 1 (before maximum circularity) with minor axis measurements from Phase 2 (after maximum circularity). Similarly, we measured the stretching axis by combining the minor axis from Phase 1 with the major axis from Phase 2 (Supplementary Fig. 2d: step 6). Finally, we generated time-evolution tables for each cell, capturing the median Brillouin shift alongside shape parameters, including apical-basal and stretching axis lengths (μm).

To plot an average time course of the cell-by-cell quantification of the Brillouin shift and the length of the apical-basal and stretching axes (Fig. 2f), we filtered those cells that acquired the largest circularity value between 00:42:20 and 01:07:44 h, corresponding to the 10th and 90th percentile of the distribution of maximum circularity times, leading to the inclusion of 7/9 cells. The maximum circularity value was measured at a median of 00:53:22 min, and this value was used in Fig. 2f to set the boundary between phase 1 and phase 2.

For the results shown in Fig. 4c, d we median-projected 3 consecutive slices of the XZ resliced volume, and generated masks for 6 cells for central mesodermal cells (with their centre on the ventral midline), for each time point from the onset of gastrulation until the mesoderm had invaginated (permeabilised embryos in PBS/control) or mesoderm cells had undergone EMT on the surface of the embryo (permeabilised embryos in 1 mg/ml Colcemid in PBS).

**Subcellular localisation of high-Brillouin shift**. To filter the largest 4% Brillouin shift values, we applied a 'default' threshold using ImageJ/Fiji on the Brillouin shift maps, using the end of cellularisation as a reference timepoint to apply the threshold.

**Quantification of the alignment of microtubules with each other.** For each EB1-GFP-expressing embryo ($N = 3$) acquired using spinning disk microscopy, we YZ re-sliced the deconvolved raw volume, and generated 5 max-projections of 20 consecutive slices each. Next, we generated rectangular masks (covering a tissue section of 73.3 μm²) that were placed within the sub-apical compartment of central mesodermal cells in each time point of each max-projected movie of each embryo. Thus, for all analysed embryos, across all timepoints, we utilised a mask of the same geometry and the same size. The rectangular masks were used to filter the EB1-GFP signal within the sub-apical compartment of central mesodermal cells.

We analysed the alignment of the filtered EB1-GFP signal with a published Matlab (R2019b) script that analyses the mean direction of the microtubule signal (either α-Tubulin immunostainings or EB1-GFP live imaging) and the dispersion of the signal around the estimated mean microtubule direction, which was termed 'MTSD'[47]. Using this Matlab script, we obtained for each embryo 5 MTSD values (one for each max-projected movie) for every analysed time point. The 5 MTSD values obtained for each timepoint were averaged, resulting in a mean MTSD and standard error of the mean per timepoint and per embryo (Fig. 4d, e).

## Statistics and reproducibility

All statistical analyses were performed in Prism Graphpad 10.2.3. Before each analysis, normality testing was evaluated using both Shapiro-Wilk test and Normal QQ plots, and based on both tests the decision to conduct parametric or non-parametric tests was made. In Fig. 1c, e, f we conducted a one-way ANOVA (parametric, matched measures). In Fig. 1h we conducted a combined linear regression model (Prism Graphpad V10.2.3) to test if the Brillouin shift dynamics in neuroectoderm and dorsal ectoderm cells can be fit with the same linear regression line. For this, we performed the combined linear regression analysis on the time course of the Brillouin shift measurements within the neuroectoderm and the dorsal ectoderm cells for each embryo between timepoints −0:04:16 h (end of cellularisation) and 01:01:23 h (ongoing squamous morphogenesis). Prism Graphpad V10.2.3 evaluates if the combined regression model yields an interaction coefficient between the neuroectoderm and dorsal ectoderm cells that is equal to 0 (zero = slope and/or intercept same for both cell types), both for the slope and the y-intercept. To analyse the correlation between the cell shape parameters (apical-basal and stretching axes length) and the Brillouin shift during the time course of squamous morphogenesis we detrended the data from the averaged cells (Fig. 2f). To proceed with the detrending, we first extracted the measurements for the time windows where there is detectable change: 23:17'–59:16' for the apical-basal cell axis length and 29:38'–57:09' for the stretching axis length. Stationarity of the time series was achieved by differencing the series once and was confirmed using the KPSS (Kwiatkowski–Phillips–Schmidt–Shin) test. An ARMA (autoregressive–moving–average) model was then fitted to each series using the 'auto.arima' function from the R package forecast. The choices of selected parameters for the models were consistent with visual inspection of the autocorrelation and partial autocorrelation plots. Spearman correlations were then computed between model residuals. The R code for this analysis is available at: https://git.embl.de/heriche/brillouin_shift_time_series

In Fig. 4e a non-parametric ANOVA Friedman test was conducted; in Fig. 4g a non-parametric Mann-Whitney test was conducted. We applied False Discovery Rate (two-stage step-up method of Benjamini, Krieger and Yekutieli; $q < 0.05$) to correct for multiple comparisons. For all statistical tests significance was set at 5% ($p < 0.05$).

No statistical method was used to predetermine sample size.

## Physical model of gastrulation

The physical model of *Drosophila* VFF was developed within the Cellular Potts (Glazier-Graner-Hogeweg) modelling framework[52], where individual cells and their internal compartments are represented by a collection of lattice sites on a fixed grid which evolves in time according to a Metropolis algorithm that minimises an effective energy that describes cell properties and behaviours (see SI for a detailed description of the model). The model was developed using the CompuCell3D (CC3D) simulation software[60], and the simulations ran on a computer cluster using a set of custom developed python scripts to efficiently explore the parameter space. The code for the simulations is publicly available at https://github.com/abhisha-ramesh/Drosophila_VFF.

The DOI for the final version of the code, for Fig. 5 and Supplementary Fig. 4 is: https://doi.org/10.5281/zenodo.15530248

The DOI for the code corresponding to Supplementary Fig. 5 is: https://doi.org/10.5281/zenodo.15530275

## Reporting summary

Further information on research design is available in the Nature Portfolio Reporting Summary linked to this article.

## Data availability

Imaging data generated in this study has been stored in Zenodo and is publicly available using the following link: https://doi.org/10.5281/zenodo.15465541

*Drosophila* fly lines used in this study are available upon request. Fly lines maintained in public repositories can also be obtained from the corresponding stock collections using the code we have provided in the 'Methods, *Drosophila* genetics'. Source data are provided with this paper.

## Code availability

The R codes developed for detrending and conducting correlation analyses, as well as codes developed for the physical model are available through the links provided in the 'Physical model of gastrulation' subsection within the Methods of this manuscript and here:

The GitHub link with the code used for the detrending methodology and associated statistical analyses is: https://git.embl.de/heriche/brillouin_shift_time_series

The DOI for the physical modelling code corresponding to Supplementary Fig. 5 is:

https://doi.org/10.5281/zenodo.15530275

The DOI for the final version of the physical modelling code, for Fig. 5 and Supplementary Fig. 4 is:

https://doi.org/10.5281/zenodo.15530248

The DOI for the physical modelling code corresponding to Supplementary Fig. 5 is:

https://doi.org/10.5281/zenodo.15530275

The code used for the analysis of the organisation of microtubules has been published in Gomez JM et al. Nat Comms 2015, and is available upon request.

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

## Acknowledgements

We thank N.H. Brown, N. Bulgakova, S. De Renzis, S. Huelsmann, K. Roeper and Y-C. Wang for reagents and fly stocks. N. Lawrence and the Gurdon Institute Imaging Facility for help with 3D-SIM imaging, EMBL Advanced Light Microscopy Facility (ALMF), in particular M. Lampe for assistance with Spinning Disk microscopy and image deconvolution, and CECAD Cologne Imaging Facility for continuous support. We acknowledge the computing resources provided by North Carolina State University High Performance Computing Services Core Facility (RRID:SCR_022168). Flybase was used throughout the project and is gratefully acknowledged. We thank E. Vogelsang for assistance with physical cross-sections of fixed embryos. We also thank C.J. Chan, N. Petridou, K. Prummel and Y.-C. Wang, for critical discussion and reading of the manuscript. Flybase was used throughout the project and is gratefully acknowledged. R.P. acknowledges support of an ERC Consolidator Grant (no. 864027, Brillouin4Life), and the German Center for Lung Research (DZL). This work was also supported by funding from the European Molecular Biology Organisation (EMBO), the University of Cologne and the Deutsche Forschungsgemeinschaft (grant DFG LE 546/12). This work was supported by the European Molecular Biology Laboratory (EMBL) and the North Carolina State University.

## Author contributions

J.M.G. conceived and designed the study as well as the analyses of processed Brillouin data and wrote the manuscript. C.B. and J.M.G. collected the data and C.B. performed the analyses of raw Brillouin data. A.T. and J.M.B. developed and implemented the in silico model. JKH performed the detrending and correlation analyses between time-evolving shape parameters and Brillouin shifts. M.L. contributed to the biological interpretation and discussion of results and aided in the selection of mutants. R.P. contributed to study and data analysis design,

and its physical interpretation, acquired funding and supervised the project. All authors provided critical feedback on all aspects of the research, and contributed to the final paper.

## Funding

## Competing interests

The authors declare no competing interests.
