## [Transparent Peer Review file · Nature Communications]

Highly dynamic mechanical transitions in embryonic cell populations during *Drosophila* gastrulation

Corresponding Author: Dr Juan Gomez

Version 0:

Reviewer comments:

Reviewer #1

(Remarks to the Author)

I read the paper "Highly dynamic mechanical transitions in embryonic cell populations during *Drosophila* gastrulation" by J. M. Gomez et al with great interest. In their work, the Authors use Brillouin microscopy to measure cell mechanical properties and their dynamics during the ventral furrow (VF) formation and invagination in the fruit fly embryo. The paper reports many interesting results and suggests the following model: During VF formation, the invaginating ventral cells progressively stiffen (cells at the ventral midline stiffen most and the peripheral mesoderm even soften a bit); lateral and dorsal cells, on the other hand, soften during VF formation. The Authors show that these changes in elastic properties correlate with cell shape changes and they attribute these dynamics to the microtubules, which--according to the Authors' hypothesis--become highly aligned due to basal movements of the nuclei, which create voids in the sub-apical cell regions. Finally, the Authors corroborate their experimental results with a Cellular Potts model, which shows that the specific dynamic changes of cells' elastic properties promote ventral furrow invagination.

Overall, this is a serious study that will be of high interest in the developmental-biology and biophysics communities. The manuscript is well written and easy to follow, the results are generally clearly presented and the manuscript, in my opinion, meets the standards of research transparency and reproducibility.

In my view, the interdisciplinary nature of this work as well as its high potential impact render this manuscript suitable for publication in Nature Communications. Below, I list a few queries that will hopefully help the Authors to strengthen their manuscript.

- 1) I find the report on Brillouin shifts in lateral and dorsal tissues a bit confusing. In particular, in p.4, the Authors conclude that "...Brillouin shift decreases at the same rate within both ectodermal cell populations during gastrulation..." What does this tell about the relative stiffnesses between the two ectodermal populations? Are these populations equally stiff? Probably not, since the experiments show that dorsal cells stretch much more than the lateral cells. So this Brillouin shift merely shows that the two cell populations undergo the same rate of softening? This is an important point that needs to be clarified.
- 2) Why did the Authors define two phases in the transition of dorsal cells from columnar to squameous? From the graphs, these two phases are not obvious to me.
- 3) As a non-biologist, I must say I do not clearly see in Fig. 4 that nuclei are moved basally. Can this be shown more clearly? Also, is it known why this kind of movement? If yes, the Authors should mention this. Particularly because the basal movement of nuclei goes against intuition. Indeed, since cells are widened at the apical side due to the natural curvature of the embryo, I would expect nuclei to be located closer to the apical side. Unless these nuclear movements happen because of apical constriction. In any case, please give some more insights on these nuclear movements.
- 4) The Authors hypothesise and that changes in cell elastic properties occur due to basal nuclear movements, which creates a void in the sub-apical cell regions, which in turn causes alignment of microtubules (MT). Can the Authors say more about how creating the sub-apical void leads to MT alignment and how does this alignment lead to stiffening? The latter is maybe obvious to some, but still this point should be explained from the mechanistic perspective.
- 5) In my opinion, the model is a very helpful addition to the paper and I see no reason why the Authors "apology" in p.10,

where they say "Although we note that the physical toy model of VFF we developed does not simulate the Brillouin scattering process..." There is no need for being "apologetic". The model contribution in this paper is clear: It connects the specific experimentally measured dynamic changes of cells' elastic properties with overall tissue deformations, which is extremely helpful.

5.1) Nevertheless, the Authors do focus mostly on the furrow depth, but they do not really comment the discrepancy of the furrow shape from experiments. Indeed the shape is significantly different, which makes me wonder about the interaction of the cells with surrounding. The Authors should explain in the main text how they account for the effects of the yolk and the vitelline membrane and what could be done in the model to improve the agreement of *in silico* cell-shape changes with those observed experimentally.

6) Font size in figures should be increased.

Most importantly:

7) In Discussion, the Authors make an attempt to reconcile their results with the results of laser microdissection experiments, published by Leptin's lab about a decade ago (Ref. [8]). This is a very important comparison, which in my opinion is missing an important point. In particular, the vertex model in Ref. [8] assumes that cell (effective) stiffness entirely originates from cell surface tensions, whereas cell interiors are incompressible. Therefore, in that model, elasticity arises due to changes of cell geometry, which in turn increase cell surface energy. If I understood well, the experiments in the Authors' paper address bulk elasticity, i.e., deformability of cell interior. But what about the contribution of surface tensions? If surface tensions are greater in lateral cell region, those cells might still be much stiffer than ventral cells.

(Remarks on code availability)

Reviewer #2

(Remarks to the Author)

This manuscript reports line-scan Brillouin microscopy imaging of the gastrulating *Drosophila* embryo, which is used to infer spatiotemporal patterns of cellular mechanical properties during mesoderm invagination. The Brillouin shifts are assumed proportional to the longitudinal cell modulus. The authors find that the sub-apical regions of ventral cells near the ventral midline exhibit increased Brillouin shift during mesoderm invagination indicating increased longitudinal modulus, whereas more peripheral ventral cells show decreased Brillouin shift. ("Sub-apical" was defined to be between the apical membrane and the apical side of the nucleus.) These patterns of Brillouin shift behavior were found to be independent of actin and myosin, but dependent on microtubules. Following treatment with the microtubule depolymerizing drug colcemid, the embryo showed reduced Brillouin shift.

A mathematical model of the *Drosophila* embryo was formulated, incorporating myosin contractile and elastic resistance forces. The model was used to examine different possible cell stiffness profiles in the ventral-lateral and apical-basal directions. Based on the model, the authors suggest tissue folding is facilitated by the dynamic patterns of cell stiffness they observe experimentally.

Major Comments

1. The conclusions presented in this work are tentative, because the reported high frequency longitudinal stiffnesses are not measurements of cell stiffness as relevant to biophysical phenomena and widely discussed in the literature. Brillouin shift (BS) measurements probe moduli at very high frequencies (GHz) whose magnitudes are typically in the gigapascal range, of order a million times greater than cell stiffnesses. The physical origins and significance of the two kinds of moduli are unrelated. It is thus very difficult to interpret BS measurements in terms of stiffness phenomena involving actin or microtubule cytoskeletal structures, whose stiffnesses are unrelated to those reported in this work.

The argument in favor of BS measurements and high frequency moduli having relevance to the much lower frequency cell stiffness phenomena of biological interest rests entirely on empirical correlations between the two, reported by some works including ref 18 cited in this manuscript: there is some evidence that when the high frequency modulus increases, the low frequency modulus increases at the same time. No mechanism for this relation is understood, to my knowledge (it seems quite possible that in some cases when one increases the other might decrease, for example). This is an important method to identify composition in cells and tissue with biomedical and other applications, and this may indirectly tell us something about large scale cellular structures, but the relation is murky at best.

2. The distant relation between the two classes of moduli is not articulated in the abstract, introduction or Results sections as far as I can see. It is dealt with in Discussion (2nd para), but this is rather late.

3. Relative changes in BS reported here are very small, typically of order 0.5 % (e.g. from 5.17 GHz to 5.18 GHz, Fig. 2a). This presumably corresponds to ~ 0.5 % relative changes in high frequency moduli. Even if one accepts an empirical relationship with cell stiffnesses, it is unclear if such tiny ~ 0.5 % changes in cell stiffness could be biologically relevant.

4. It is argued that the change in BS is attributable to microtubules, based on colcemid-treated embryos that exhibit a lower BS than control (Fig. 4f). However, the furrow becomes more open; it is possible the more open furrow directly alters the BS, rather than a direct effect from microtubules (i.e. any change that alters the furrow shape, microtubule-related or other, may

conceivably affect the BS similarly).

5. Throughout the mathematical model section the “optimal” furrow formation is referred to as a standard against which to compare model results. However no experimental evidence is cited for furrow depth or shape to justify the “optimal” definition.

Minor issues:

1. Fig. 1h shows linear regression results for the Brillouin shifts of the dorsal ectoderm and neuroectoderm, to demonstrate they have the same slope. However, to demonstrate this needs two distinct regression fits, with a statistical test to compare the two fitted slopes.

2. Fig. 2g shows Pearson correlations between Brillouin shift and the lengths in the apical-basal and stretching axes, respectively, from Fig. 2f. Since in Fig. 2f the curves have strong trends in common, I believe that measuring the Pearson correlation is not appropriate for this situation.

3. Last paragraph of “The role of microtubules in the determination of mechanical properties during VFF.” Fig. 4e is referred to for results on twist mutant embryos, but as far as I can see Fig. 4e does not involve twist mutants.

(Remarks on code availability)

Reviewer #3

(Remarks to the Author)

Gomez et al. describe the experimental characterization of embryonic cells during *Drosophila* gastrulation using line-scan Brillouin microscopy. The Brillouin frequency shifts exhibited spatial and temporal variations on a time scale of minutes during morphogenesis. These changes were compared with the distribution and concentration of several structural molecules, such as actin fibers and microtubules. While the Brillouin data presented are technically sound, novel, and interesting, the interpretation of these images in terms of cell stiffness, and their connection to structural molecules, requires more careful consideration and supporting data.

As the authors mention, Brillouin microscopy measures a specific mechanical property, namely the longitudinal modulus (both real and imaginary components). Throughout the manuscript, the Brillouin frequency shifts are often referred to as “stiffness” or “mechanical properties,” terms also used in the context of AFM and other mechanical measurements from the literature, which typically refer to “shear modulus” or “Young’s modulus.” However, the longitudinal modulus and shear modulus are distinct properties. Although they can be correlated in biological tissues, this relationship is phenomenological rather than fundamental. This distinction is insufficiently clarified in the manuscript. For example, in the very first sentence of the abstract, the mechanical properties that affect cell morphology in response to cellular forces mostly pertain to shear moduli. The role of longitudinal moduli in determining cell shape is thought to be minimal, if not negligible.

The authors appear to assume that higher Brillouin frequencies correspond to higher stiffness and attempt to interpret the data using established knowledge around shear stiffness, such as the relationship between cell deformability and structural molecules. However, since the correlation between longitudinal and shear moduli is not guaranteed and can be influenced by other physical properties—particularly the solid-to-fluid ratio—this approach risks leading to erroneous conclusions. Therefore, my primary concern is the accuracy of the physical model presented.

Even if a quantitative correlation between longitudinal and shear moduli were to exist in these embryos during morphogenesis, it could be phenomenological rather than mechanistic. For instance, intracellular protein concentration might increase alongside microtubule assembly, leading to decreased water content and a higher solid-to-fluid ratio, which would then increase the Brillouin frequency. In this case, shear stiffening by microtubules may not be the direct cause of the longitudinal modulus increase.

That said, the results clearly demonstrate that the longitudinal modulus of cells changes dynamically and spatiotemporally, which points to the potential utility of the technique for identifying such changes.

In conclusion, while the technological advance enabling time-lapse imaging of whole organisms is impressive, the interpretation of the data in terms of cell deformation and structural molecules needs to be addressed more thoroughly and clearly.

(Remarks on code availability)

Version 1:

Reviewer comments:

Reviewer #1

(Remarks to the Author)

I read the revised manuscript by Gomez et al. The Authors have responded to initial Reviewers' reports and edited the manuscript accordingly. I do not have any further queries and I recommend the paper for publication in Nat. Commun.

(Remarks on code availability)

Reviewer #2

(Remarks to the Author)

In the main, the authors have responded satisfactorily to the major points raised in my earlier review.

I suggest they add to the abstract a mention of the vast discrepancy (at least six orders of magnitude) between the timescales probed by this experimental method, and the biological timescales of interest to this study. This is a far more important issue than the particular modulus involved (longitudinal vs shear etc). The revised submission now states this fact in the Introduction, but the key sentence in the abstract reads "... we characterize and spatially map in three dimensions the dynamics of the longitudinal modulus as a readout for the evolving cell material properties.... utilizing line-scan Brillouin microscopy." Most biological researchers who read this sentence will likely be unfamiliar with the method, and would therefore naturally interpret "the modulus" as referring to the biophysically and biologically relevant modulus. In fact, the sentence refers to a remote quantity characterizing much higher frequency behavior, which some correlative data suggests may be linked to the much lower frequency modulus of biological interest.

(Remarks on code availability)

Reviewer #3

(Remarks to the Author)

The revised manuscript is much improved, offering more thorough and accurate interpretations of the experimental data.

(Remarks on code availability)

response to reviewers' comments

Dear Reviewers,

We sincerely appreciate your thoughtful feedback, which has been instrumental in strengthening our manuscript. In response, we have made extensive revisions, including refining the rheological definitions, clarifying the background and limitations of Brillouin microscopy, and enhancing the discussion of its scope and relationship to complementary technologies (e.g., AFM). Additionally, we have ensured a more precise use of the mechanical terminology and carefully moderated our conclusions to provide a more balanced and rigorous interpretation of our findings.

Additionally, several of your comments raised particularly insightful points, which we have incorporated into the Discussion section. Given the overlap in concerns raised by Reviewers 2 and 3 regarding the nature of Brillouin microscopy measurements, we have provided a consolidated response before addressing their individual comments.

We hope you find our revised manuscript significantly improved and now suitable for publication in *Nature Communications*.

Yours sincerely,

Dr. Juan Manuel Gómez

Dr. Robert Prevedel

Reviewer #1 (Remarks to the Author)

General comments

I read the paper "Highly dynamic mechanical transitions in embryonic cell populations during *Drosophila* gastrulation" by J. M. Gomez et al with great interest. In their work, the Authors use Brillouin microscopy to measure cell mechanical properties and their dynamics during the ventral furrow (VF) formation and invagination in the fruit fly embryo. The paper reports many interesting results and suggests the following model: During VF formation, the invaginating ventral cells progressively stiffen (cells at the ventral midline stiffen most and the peripheral mesoderm even soften a bit); lateral and dorsal cells, on the other hand, soften during VF formation. The Authors show that these changes in elastic properties correlate with cell shape changes and they attribute these dynamics to the microtubules, which--according to the Authors' hypothesis--become highly aligned due to basal movements of the nuclei, which create voids in the sub-apical cell regions. Finally, the Authors corroborate their experimental results with a Cellular Potts model, which shows that the specific dynamic changes of cells' elastic properties promote ventral furrow invagination.

Overall, this is a serious study that will be of high interest in the developmental-biology and biophysics communities. The manuscript is well written and easy to follow, the results are generally clearly presented and the manuscripts, in my opinion, meets the standards of research transparency and reproducibility.

In my view, the interdisciplinary nature of this work as well as its high potential impact render this manuscript suitable for publication in Nature Communications. Below, I list a few queries that will hopefully help the Authors to strengthen their manuscript.

1. I find the report on Brillouin shifts in lateral and dorsal tissues a bit confusing. In particular, in p.4, the Authors conclude that "...Brillouin shift decreases at the same rate within both ectodermal cell populations during gastrulation..." What does this tell about the relative stiffnesses between the two ectodermal populations? Are these populations equally stiff?

Probably not, since the experiments show that dorsal cells stretch much more than the lateral cells. So this Brillouin shift merely shows that the two cell populations undergo the same rate of softening? This is an important point that needs to be clarified.

The Brillouin shift indeed decreases at the same rate in both ectodermal cell populations, the neuroectoderm and the ectoderm, suggesting a similar rate of softening. But furthermore, the shift also decreases in a comparable manner when both ectodermal cell populations are undergoing their corresponding morphogenetic event. This can be observed in Fig. 1h, and in the new panels in **Supplementary Fig. 2a,b**. This observation guided us to test whether the dynamics of the Brillouin shift in both cell populations could be explained with a combined/general linear regression model. The statistical test confirmed that one line can explain the dynamics of both ectodermal cell populations, even though dorsal ectodermal cells then further reduce their Brillouin shift until the end of squamous morphogenesis. Thus, the neuroectoderm and the ectoderm exhibit

remarkably similar stiffness values at each time point where Brillouin shifts were measured for both tissues.

Action

To clarify this point we have added new panels in **Supplementary Fig 2a,b** and included in the Results section the following paragraph:

(...)Linear regression analyses of neuroectoderm and dorsal ectoderm cells suggested that the Brillouin shift decreases in a comparable manner in both populations during gastrulation (Supplementary Fig. 2a,b), with the dorsal ectoderm undergoing further reduction in the Brillouin shift -compared to the neuroectoderm- until completion of squamous morphogenesis (Fig. 1e,g). To study whether their Brillouin shift dynamics were truly comparable, we tested if the individual regressions for the neuroectoderm and ectoderm could be described by a combined linear regression model. This showed that the evolution over time of the Brillouin shift in both cell types can be fit by the same line (Brillouin Shift (MHz) = $-0.33 \times t + 0.28$; slope comparison: $p=0.084$; intercept comparison: $p=0.795$). These results support a model in which cells along the dorsal-ventral axis exhibit two types of material properties: biphasic Brillouin shift behavior in the mesoderm and softening at similar rates across ectodermal cells.(...)

2. Why did the Authors define two phases in the transition of dorsal cells from columnar to squameous? From the graphs, these two phases are not obvious to me.

We defined the two phases for operational reasons based on the morphology of the cells, and we did this only to make our measurements simpler. During the transition from columnar to squamous, the cells go through a point where the two axes are equal in length, i.e. the cell has an approximately circular cross section (Supplementary Fig. 2c,d). This makes it easier to define the apical-basal (shortening) and circumferential (stretching) axes as the lengths of the major and minor cell axes before the largest circular point, and the reverse after this point (Supplementary Fig. 2d).

Action

We have now improved the panels in **Supplementary Fig. 2c,d** to clearly indicate the swap of what constitutes the major cell axis, improved the explanation of the definition and utility of defining these *arbitrary* phases in the Results section:

(...)When cells transition from a columnar to a squamous shape, their longest (i.e., major) axis is initially the apical-basal axis, whereas the shortest (i.e., minor) is circumferential and along the dorso-ventral direction in which the cell stretches (i.e., 'stretching axis'; Supplementary Fig. 2c). As the cell shortens along the apical-basal axis, it reaches a point where the apical-basal axis length is comparable to the stretching axis (Supplementary Fig. 2c,d). At this point the fitting ellipses become roughly circular (detected using the circularity value; Supplementary Fig. 2d, see Methods), and the major and minor geometric axes of the cell change relative to the directions in the cell. Therefore, we used this point with maximum circularity to operationally divide the process into two phases (Supplementary fig. 2c,d).(...)

and in the Methods section:

(...)At the onset of squamous morphogenesis, the apical-basal axis aligns with the major axis of the fitted ellipse, while the perpendicular 'stretching axis' corresponds to the minor axis. As cells adopt an approximately isotropic shape, these axes switch: the major axis becomes the 'stretching axis,' and the minor axis becomes the apical-basal

axis. We identified this transition by measuring the 'circularity' of the fitted ellipses and divided the process into two phases: Phase 1, from the initiation of squamous morphogenesis until cells reach isotropy, and Phase 2, from isotropy to the completion of morphogenesis (Supplementary Fig. 2d: step 3). The time point at which the fitted ellipses reach maximum circularity marks the transition from Phase 1 to Phase 2 (Supplementary Fig. 2d: steps 4-5). To track the apical-basal axis in each cell, we combined major axis measurements from Phase 1 (before maximum circularity) with minor axis measurements from Phase 2 (after maximum circularity). Similarly, we measured the stretching axis by combining the minor axis from Phase 1 with the major axis from Phase 2 (Supplementary Fig. 2d: step 6). Finally, we generated time-evolution tables for each cell, capturing the median Brillouin shift alongside shape parameters, including apical-basal and stretching axis lengths (μm).(...)

3. As a non-biologist, I must say I do not clearly see in Fig. 4 that nuclei are moved basally. Can this be shown more clearly? Also, is it known why this kind of movement? If yes, the Authors should mention this. Particularly because the basal movement of nuclei goes against intuition. Indeed, since cells are widened at the apical side due to the natural curvature of the embryo, I would expect nuclei to be located closer to the apical side. Unless these nuclear movements happen because of apical constriction. In any case, please give some more insights on these nuclear movements.

It is indeed the case that the nuclei are initially positioned at the widest part of the cell, i.e. at the apical end. The basal displacement of the nuclei was described during the early characterisation of the cellular events that lead to the formation of the ventral furrow (Sweeton D. et al, *Development*, 1991; Kam Z. et al, *Cell Reports*, 1991). The mechanisms driving the basal displacement of the nuclei were investigated in the following years, with two main outcomes, an association between the hydrodynamic basal flows and the basal displacement of nuclei, and the requirement of microtubules (Gomez JM et al., *eLife*, 2024) for the positioning and basal displacement of nuclei during ventral furrow formation.

Action

To improve the clarity of the basal displacement of the nuclei, we have added a panel showing the initial apical position of nuclei, and their basal relocalisation during ventral furrow formation in **Supplementary Figure 3a**, incorporated additional references from the abovementioned studies, and added more background in the Results section:

'(...)At the onset of VFF, the nuclei are located in the apical end of the cell. However, when apical constriction starts, the nuclei move basally (Sweeton D. et al. *Development* 1991; Kam Z. et al. *Development*; Gomez JM et al. *eLife* 2024; Supplementary Fig. 3a: Nuclei labelled with Snail transcription factor, inset: blue dashed-line) together with the centrosomes (labelled with the centriole component Asterless [33]; Fig. 4a, arrows), a movement that has been associated with the basal hydrodynamic flow of the cytoplasm (Gelbart M. et al. *PNAS* 2012; He B. et al. *Nature* 2014) and that requires microtubules (Gomez JM. et al. *eLife* 2024). The movement of the centrosomes and the nuclei towards the basal side of cells increases the distance between the apical membrane and the nuclei, generating a longer subapical compartment in central mesodermal cells (Fig. 4a, insets, Supplementary Fig. 3a)(...).'

4. The Authors hypothesise that changes in cell elastic properties occur due to basal nuclear movements, which creates a void in the sub-apical cell regions, which in turn causes alignment of microtubules (MT). Can the Authors say more about how creating the sub-apical void leads to MT alignment(...)?

The organisation of microtubules is by default dictated by cell shape in epithelial tissues. In cells with anisotropic geometry, microtubules naturally align with each other along the cell's main axis. A previous study by some of us (Gomez JM et al. NComms 2016) identified this default mechanism of microtubule organisation. In particular, the increased alignment of microtubules is observed as cell elongation (or its anisotropy) increases. Therefore, this alignment mechanism does not necessitate the generation of a 'void' compartment *per se*. Instead it relies on an increase in the anisotropy of the sub-apical compartment, defined between the apical cortex and the apical side of the nuclei. When nuclei move basally during apical constriction, the sub-apical compartment increases its length along the apical-basal axis, allowing microtubules to polymerise preferentially along the apical-basal direction.

(...) and how does this alignment lead to stiffening? The latter is maybe obvious to some, but still this point should be explained from the mechanistic perspective.

To understand how the global reorganisation of cellular polymers translates into an increase in the Brillouin shift, and consequently into stiffening, it is important to emphasise that the longitudinal modulus is measured along a specific direction (corresponding to the optical axis of the detection objective in our case). Microtubules are rigid polymers with a greater persistence length than F-actin (Gittes F. et al., JCB 1993). Intuitively, microtubules can be thought of as rigid fibres that, when aligned, make the biological material significantly more resistant to deformation along the fibre's axis compared to perpendicular directions. This leverages the relevance of probing the mechanical properties in a direction-sensitive manner, as Brillouin microscopy does. In the particular morphogenetic event we studied, microtubules become on average aligned with each other and with the longer cell axis, the cell's apical-basal axis (Fig 4c-e, Supplementary Video 11) -as measured with the MTSD parameter (Gomez JM et al., NComms, 2016). When microtubules become aligned with each other and with the cell's apical-basal axis, they also become highly aligned with the probing direction, potentially contributing to an increased Brillouin shift, and thus, an increased longitudinal modulus along the same direction.

A clear demonstration of this directional sensitivity in Brillouin microscopy was reported in studies of the extracellular matrix (ECM) in the zebrafish notochord (Palombo F. et al., J.R. Soc. Interface, 2014; Bevilacqua C. et al., Biomed. Opt. Express, 2019). The ECM in this tissue consists of strongly aligned fibres, and the Brillouin shift was found to be markedly larger when the probing direction was parallel to these fibres than when perpendicular, underscoring Brillouin microscopy's sensitivity to fibre alignment. Additional examples of this directional sensitivity have been reported in plants (Elsayad K. et al. Sci Signalling 2016), viscose fibers (Czibula C. et al. J. Phys. Photonics) and other biological materials such as spider silk and shells (Koski K. Nat. Material 2012, Radhakrishnan D. et al. ACS Applied BioMaterials).

Action

To address this point in the revised manuscript, we have introduced an improved explanation about how cell shape directs microtubule alignment within the Results section:

'(...)This enhanced alignment of sub-apical MTs relative to each other and to the apical-basal axis in central mesodermal cells is consistent with the reported cell shape-driven mechanism for MT organisation, where MTs

become more aligned as the cellular compartment in which they grow becomes increasingly elongated (Gomez JM et al. Nat Comms 2015).(...)

and how microtubule alignment may lead to an increase in the Brillouin shift within the Results section:

(...)We reasoned that when sub-apical MTs align parallel to the apical-basal axis, they may increase the Brillouin shift of mesoderm cells along the longitudinal direction, because these are stiff fibers (Gittes F. et al. JCB 1993). The alignment of sub-apical microtubules along the cell's apical-basal is parallel to the probing direction of the Brillouin microscope, and thus, is consistent with the measured increase in Brillouin shift during mesoderm invagination. Similar results have been previously obtained for ECM fibers *ex vivo* (Palombo F. et al. J. R. Soc. Interface 2014) and *in vivo* (Bevilacqua C. et al. Biomed. Opt. Express 2019). Consistent with this, the alignment of sub-apical MTs with the apical-basal cellular axis coincides with the time when we measure the peak in Brillouin shift (Fig. 1b,c; Fig. 4c-e; Supplementary Video 3,11).(...)

We have also rewritten the following paragraph in the Discussion section to clarify this point:

(...)We propose a model in which the observed changes in the material properties of the mesoderm along the longitudinal direction arise from the reorganisation of microtubules. In epithelial tissues, microtubule organisation is dictated by the degree of cell elongation, with microtubules becoming increasingly aligned as cells elongate(Gomez JM et al. Nat Comms 2016). We speculate that a similar mechanism regulates the alignment of sub-apical microtubules as the distance between the nuclei and the apical surface increases during VFF, leading to the elongation of the sub-apical compartment (Supplementary Fig. 3a). This process could facilitate microtubule alignment along the apical-basal axis, filling the sub-apical compartment with stiff polymers(Gittes F. et al. JCB 1993). Because Brillouin microscopy measures mechanical properties directionally, parallel to the apical-basal axis in this case, the increasingly aligned microtubules within the sub-apical compartment (Fig. 4c–e) are strong candidates for dynamically stiffening mesodermal cells, thereby promoting fold formation. In summary, our results suggest that microtubule reorganisation is ultimately driven by apical constriction and the elongation of central mesodermal cells, a cost-effective mechanism that has also been observed in the *Drosophila* epidermis later in development (Gomez JM et al. Nat Comms 2016).(...)

5. In my opinion, the model is a very helpful addition to the paper and I see no reason why the Authors "apology" in p.10, where they say "Although we note that the physical toy model of VFF we developed does not simulate the Brillouin scattering process..." There is no need for being "apologetic". The model contribution in this paper is clear: It connects the specific experimentally measured dynamic changes of cells' elastic properties with overall tissue deformations, which is extremely helpful.

We thank the Reviewer for this encouraging comment.

Action

We have now rephrased these sentences in the Discussion section to improve clarity and be less 'apologetic' of the model rationale and usefulness for understanding the material properties dynamics during folding events.

'(...)Our physical model of ventral furrow formation supports a role for the longitudinal modulus, which is derived from our Brillouin measurements. Additionally, the physical model shows that dynamic and localised changes in cellular material properties -which so far have been assumed to be constant during a morphological process-, promote

the progress of folding events, even though the model only utilizes our measurements in a qualitative way and does not physically simulate the high-frequency aspects of the Brillouin scattering process itself. (...)

6. Nevertheless, the Authors do focus mostly on the furrow depth, but they do not really comment on the discrepancy of the furrow shape from experiments. Indeed the shape is significantly different, which makes me wonder about the interaction of the cells with surrounding. The Authors should explain in the main text how they account for the effects of the yolk and the vitelline membrane and what could be done in the model to improve the agreement of *in silico* cell-shape changes with those observed experimentally.

We agree with the Reviewer's comments about the discrepancy of the overall shape of the cells and tissue by the end of the simulations. This happens because in this first version of our physical model we did not incorporate cell lengthening along the cell's apical-basal direction that occurs at the onset of gastrulation, due to the cellularisation process, and afterwards, due to the apical constriction (Gelbart M. et al PNAS 2012) .

Action

To address this we improved our model to add a second core compartment (core2) situated between the geometric center (i.e. original core) and the basal domains of the cells. This compartment increases over time to match/replicate the cell's apical-basal lengthening (Fig. 5b). We also added a vitelline membrane surrounding the cells to guide the growth/lengthening of the cells inward.

The addition of this process greatly improved the correspondence of our simulated cells and tissue shapes with the real embryo, and also, with the mechanical characterisation using Brillouin microscopy. We note that in the new model the stiffness of the central cells becomes more relevant to furrow depth than the stiffness of the peripheral cells (see new Fig. 5e), which is in accordance with our experimental results that indicate an increased Brillouin shift in the central cells. At the same time, when stiffness is allowed to change dynamically in the model, we note that the region of improved (increased) furrow depth is expanded towards higher reductions in the stiffness of the peripheral cells (see new Fig. 5f), again in accordance with our observed reduction in Brillouin shift for these cell populations. Other important results such as the difference in the roles of sub-basal vs sub-apical stiffness (Fig. 5d), and the advantage of a dynamic stiffness vs a static stiffness profile (Supplemental Fig. 3d) remains the same. We also added more results about the static vs dynamic stiffness for other stiffness starting values to show the validity of these results over a wide range of model parameters (Supplementary Fig 4e-h).

We added a new supplemental figure (Supplementary Fig. 5) with the results of the old model without cell growth/heightening and discussed in the Supplemental Information how most of the qualitative results are independent of this additional process, but are improved when we add cell growth/heightening in the new model.

7. Font size in figures should be increased.

Font sizes in figures have been increased.

Most importantly:

8. In Discussion, the Authors make an attempt to reconcile their results with the results of laser microdissection experiments, published by Leptin's lab about a decade ago (Ref. [8]). This is a very important comparison, which in my opinion is missing an important point. In particular, the vertex model in Ref. [8] assumes that cell (effective) stiffness entirely originates from cell surface tensions, whereas cell interiors are incompressible. Therefore, in that model, elasticity arises due to changes of cell geometry, which in turn increase cell surface energy. If I understood well, the experiments in the Authors' paper address bulk elasticity, i.e., deformability of cell interior. But what about the contribution of surface tensions? If surface tensions are greater in lateral cell region, those cells might still be much stiffer than ventral cells.

The Reviewer raises an important point in highlighting the differences between our work and the study by Rauzi M. et al. which, to our knowledge, is the most complete precedent for the mechanical characterisation of dorsal-ventral cell populations at gastrulation stage.

In this study, laser microdissection experiments were conducted on the apical side of cells, thereby probing -specifically- the tension fields along the apical cortex in a direction tangential to the cell surface. Using Brillouin microscopy, measurements were conducted orthogonal to apical tension fields, along the apical-basal axis.

Using Brillouin microscopy, we characterise the longitudinal modulus within the whole cell volume with a resolution given by our microscope's point-spread function, which is on the order of $3\mu\text{m}$ in the axial direction. This unfortunately does not allow us to resolve the cellular membrane and its associated actin meshwork, the cortex. Both at the apical and lateral sides of the cell, the Brillouin shift represents an average between the components of the corresponding cell border domain (membrane, cortex and associated proteins) and either the perivitelline medium -apically- or the cell interior -laterally-. Therefore, despite having subcellular resolution, we are not able to uniquely resolve the cell boundary (or surface) of each blastoderm cell.

To summarise, laser microdissection and Brillouin microscopy probed different mechanics, in directions orthogonal to each other and in different cellular regions, complementing each other. We consider it is indeed plausible that the apical surface of neuroectoderm cells is stiffer compared to that of ventral and dorsal cells while they soften along the longitudinal direction (apical-basal axis). This highlights the importance of combining methodologies to characterise mechanical transitions at the tissue and organismal level.

Action

We have now expanded the discussion paragraph that compares the study by Rauzi et al., including sentences that clarify this point raised by the Reviewer:

(...) The most complete mechanical characterisation of dorsal-ventral cells that preceded our work was conducted by Rauzi M. et al. (2015). In this study, the authors characterised the mechanics of dorso-ventral cells at gastrulation stage using laser microdissection along the apical cell surface. Assuming cellular mechanics arise from differential cell surface tensions, the authors conclude that ectodermal and mesodermal cells soften whereas neuroectodermal cells stiffen during gastrulation (Rauzi M. et al Nat Comms 2015). How can this be integrated with our results? Laser microdissection measures the cortical

tensions (i.e., a force) in a direction tangential to the apical surface of the epithelium, potentially averaging contributions from a larger field of cells. In contrast, Brillouin microscopy measures material properties orthogonally to the apical surface and across the entire cell volume with a sub-cellular resolution determined by the axial size of the point-spread function, which is in the order of $3\mu\text{m}$. Therefore, we are unable to specifically resolve the mechanics of the cellular membrane and its associated actin meshwork, in contrast to laser microdissection analyses. An additional difference between both studies is the time-resolution: whereas laser microdissection can be performed only once and in few isolated places, using Brillouin microscopy we capture and spatially map the dynamics of whole cell material properties in the same specimen at regular timepoints. We therefore consider it is indeed plausible that the apical surface of neuroectoderm cells is stiffer compared to that of ventral and dorsal cells while they soften along the longitudinal direction (apical-basal axis). This highlights -again- the importance of combining methodologies to characterise mechanical transitions at the tissue and organismal level.(...)

Shared Response for Reviewers 2 and 3:

The Reviewers raise interesting and important points. Before responding to their specific comments, we would like to discuss a broader perspective on Brillouin microscopy, address common themes, and clarify key points by outlining the fundamental principles of both Brillouin scattering and mechanobiology.

First, we would like to clarify aspects of the terminology we use in the manuscript. As Reviewers 2 and 3 note, in biological sciences, the term 'stiffness' has traditionally been associated with moduli other than the longitudinal modulus, for reasons we further elaborate in the following paragraph. However, we wish to emphasise that the term '*stiffness*' is by all means not reserved for a single type of modulus with specific boundary conditions measured at a given frequency. From a physical point of view, *stiffness* is represented by the full elasticity tensor, which is inherently dependent on both the probing frequency as well as the wavevector (directionality). Even in an isotropic, homogeneous, and dispersionless material, the elasticity tensor comprises multiple independent components, encompassing tensile, shear, and longitudinal moduli, each associated with different stress and strain directions. Nonetheless, to prevent any ambiguity regarding the interpretation of stiffness in this study, we have refrained from using this term, except in relation to the physical model, where we employ the term *stiffness* and thus, explicitly define it (see Supplementary Information, Supplementary Note 1). Instead, we now refer to the changes in the material properties as changes in the Brillouin shift or the longitudinal modulus throughout the text.

Since it is practically not feasible to measure the full elastic tensor in most biological systems, most studies focus on a single modulus, thus defining the specific stress-strain direction that they are probing. In the case of Brillouin measurements, the longitudinal modulus corresponds to a stress-strain constrained longitudinally along the optical axis. The morphogenetic events that occur in dorso-ventral cells during gastrulation involve changes in the shape of cells along their apical-basal axis (or longitudinal direction, aligned with the optical axis of our microscope), both for mesodermal and ectodermal cells, thus making the longitudinal modulus relevant and suitable to characterise the mechanics of this morphogenetic process.

The longitudinal modulus is less well characterised in living systems because it is typically not accessible by standard and more widespread techniques. In fact most mechanobiological studies still rely on established methodologies such as atomic force microscopy (AFM),

micropipette aspiration, and ferrofluids, which typically measure Young's and shear moduli at low frequencies (see e.g. Review by Petridou N. and Heisenberg CP EMBO J 2019), as both Reviewers noted. These methods have enabled extensive characterisation of these moduli in living systems. However, we would like to stress again that there is no theoretical nor fundamental basis to exclude the longitudinal modulus. Additionally, Brillouin microscopy offers significant advantages to perform non-invasive measurements of living samples. This is especially valuable at the organismal level where accessibility poses a challenge. For example, in *Drosophila* embryos, the blastoderm is enclosed by two protective layers: the chorion, which can be removed while keeping the embryo viable, and the vitelline membrane, which cannot be removed without compromising embryo viability. The vitelline membrane, approximately 300 nm thick (Margaritis LH, Kafatos FC, Petri WH. J Cell Sci. 1980), further limits access to the blastoderm. Attempts to measure blastoderm mechanics using AFM (in collaboration with Alba Diz-Muñoz -EMBL Heidelberg) were constrained by the presence of this membrane. Similarly, efforts by co-authors JM Gomez and M Leptin to use ferrofluids (Serwane F, Mongera A et al. Nat Methods 2017) to measure subcellular mechanical dynamics in *Drosophila* embryos were hindered by the inability to consistently generate subcellular-sized droplets, rendering the method unsuitable for our purposes. In contrast, Brillouin microscopy provides a non-invasive, high-resolution alternative capable of probing intracellular mechanics over time in the same sample/specimen without the limitations of surface-restricted measurements or droplet generation. While the high-frequency, longitudinal modulus is still relatively new in mechanobiology, we believe that Brillouin microscopy's unique advantages outweigh its limitations, making it the most suitable tool for addressing questions during fast morphogenetic events. **We elaborate further about this in points 9-12 and 18-22.**

In response to many of the points raised, we have rewritten several sections of the manuscript during our revision. Importantly, we have elaborated on the definition of the longitudinal modulus and clarified the differences to other mechanical moduli used in the field. We now also clearly state that Brillouin microscopy measures material properties at high frequencies, and how high-frequency and low-frequency measurements can be related. In addition, we have further highlighted the advantages of using Brillouin microscopy instead of other established methodologies for probing mechanical properties, such as AFM and ferrofluids, in the context of developmental biology. Finally, we have revised our conclusions, which now includes a broader discussion of the characteristics and limitations of Brillouin microscopy.

Reviewer #2 (Remarks to the Author):

This manuscript reports line-scan Brillouin microscopy imaging of the gastrulating *Drosophila* embryo, which is used to infer spatiotemporal patterns of cellular mechanical properties during mesoderm invagination. The Brillouin shifts are assumed proportional to the longitudinal cell modulus. The authors find that the sub-apical regions of ventral cells near the ventral midline exhibit increased Brillouin shift during mesoderm invagination indicating increased longitudinal modulus, whereas more peripheral ventral cells show decreased Brillouin shift. ("Sub-apical" was defined to be between the apical membrane and the apical side of the nucleus.) These patterns of Brillouin shift behavior were found to be independent of actin and myosin, but dependent on

microtubules. Following treatment with the microtubule depolymerizing drug colcemid, the embryo showed reduced Brillouin shift.

A mathematical model of the *Drosophila* embryo was formulated, incorporating myosin contractile and elastic resistance forces. The model was used to examine different possible cell stiffness profiles in the ventral-lateral and apical-basal directions. Based on the model, the authors suggest tissue folding is facilitated by the dynamic patterns of cell stiffness they observe experimentally.

Major Comments

9. The conclusions presented in this work are tentative, because the reported high frequency longitudinal stiffnesses are not measurements of cell stiffness as relevant to biophysical phenomena and widely discussed in the literature. Brillouin shift (BS) measurements probe moduli at very high frequencies (GHz) whose magnitudes are typically in the gigapascal range, of order a million times greater than cell stiffnesses. The physical origins and significance of the two kinds of moduli are unrelated. It is thus very difficult to interpret BS measurements in terms of stiffness phenomena involving actin or microtubule cytoskeletal structures, whose stiffnesses are unrelated to those reported in this work.

We thank the Reviewer for bringing up an important issue that we have failed to properly explain and put in context in the original manuscript. The Reviewer is correct in noting that BM measures the longitudinal modulus at high-frequency. However, as we have already stated in the general response, we disagree with the statement of the Reviewer that the longitudinal modulus is not relevant to biophysical phenomena. There is no physical reason why the longitudinal modulus should not be representative (and important) for biophysical measurements of cell stiffness. In fact, -as mentioned in the general response- morphogenetic events that occur in dorso-ventral cells during gastrulation involve changes in the shape of cells along their apical-basal axis (or longitudinal direction), both for mesodermal and ectodermal cells, thus making the longitudinal modulus relevant and suitable to characterise the mechanics of this morphogenetic process. As mentioned in the original manuscript, while the longitudinal modulus is fundamentally different from the other moduli that constitute the elastic tensor, it is still a valuable mechanical measurement that has been validated in various physical and biological settings(Scarcelli G. et al. Nat Methods 2015; Elsayad K. et al. Sci Signal 2016; Palombo F. et al. Chem Rev 2019; Prevedel R. et al. Nat Methods 2019; Schlüßler R. et al. Biophysical Journal 2018).

Indeed, the physical origin and frequency scale of the Brillouin measurements is different from the typically used moduli in mechanobiology. We will expand about this subject in the following point (point 10).

We fully acknowledge that applications of Brillouin microscopy and its measurements are relatively new in biology, and therefore the technique, its limitations and relation to more established techniques need to be properly introduced and clarified.

Action

We have therefore revised the introduction and discussion section, and specifically refer to the peculiarities of Brillouin microscopy where appropriate - this should also address comments of the Reviewer further below. Among others, we now elaborate on the frequency differences between the moduli and measurements:

(...) The shift in energy of the Brillouin-scattered light is related to the sound velocity of the probed material, which is dependent on its material properties (Prevedel R. et al. Nat Methods 2019; Bouvet P. et al. BiorXiv 2024). To characterise the material properties, the full elasticity tensor is required, which is inherently dependent on both the probing frequency as well as the wavevector (directionality). Even in an isotropic material, the elasticity tensor comprises multiple independent components, encompassing tensile, shear, and longitudinal moduli, each associated with different stress and strain directions. The longitudinal modulus, defined as the ratio between the uniaxial stress to strain, can be calculated from the Brillouin shift if the refractive index and mass density are known. However, even in the absence of these parameters, the Brillouin spectrum can be used as a proxy of visco-elastic properties (Schüßler R. et al. Biophysical J 2018). Indeed in this study we report the Brillouin shift as a proxy to the longitudinal modulus, as commonly done in the field.(...)

and:

(...) Brillouin microscopy differs from more established rheological methods in two main aspects. First, Brillouin microscopy measures the longitudinal modulus whereas established rheological methods such as AFM measure the Young's and/or Shear Moduli (Prevedel R. et al Nat Methods 2019; Petridou N. et al. EMBO J 2019). Because the longitudinal modulus is typically not accessible by standard techniques (e.g. AFM), it has been less characterised in biological systems. Notably, the Young's, shear and longitudinal moduli describe different responses of materials to particular stresses. They can all be derived from the elasticity tensor (Bouvet P. et al. BiorXiv 2024) but measuring all the independent components is practically not feasible in anisotropic materials such as many biological systems. Second, Brillouin microscopy measures mechanical properties at GHz frequencies, probing material responses on timescales several orders of magnitude smaller (~nanoseconds) than those of cellular-level biological processes (~milliseconds). Despite this, direct comparisons between the two techniques show high empirical correlations between these two regimes (Scarcelli G. et al. 2011 Biophysical J.; Scarcelli G. et al. Nat Methods 2015; Weber I. et al Phys Biology 2017; Zhang J. et al. Nat Methods 2023).(...)

and:

(...) While its limitations and assumptions must be carefully considered, Brillouin microscopy provides a powerful platform for exploring the mechanobiology of developmental processes with subcellular resolution, providing insightful information about the material properties of (sub-) as well as cellular compartments. The utility of Brillouin microscopy is particularly evident in scenarios where the target tissue is inaccessible to other contact-based techniques, such as AFM. This is exemplified in studies of *Drosophila* gastrulating embryos, where the blastoderm is encased by the vitelline membrane, rendering direct mechanical access and measurements of the underlying blastoderm unfeasible.(...)

10. The argument in favor of BS measurements and high frequency moduli having relevance to the much lower frequency cell stiffness phenomena of biological interest rests entirely on empirical correlations between the two, reported by some works including ref 18 cited in this manuscript: there is some evidence that when the high frequency modulus increases, the low frequency modulus increases at the same time. No mechanism for this relation is understood, to my knowledge (it seems quite possible that in some cases when one increases the other might decrease, for example). This is an important method to identify composition in cells and tissue

with biomedical and other applications, and this may indirectly tell us something about large scale cellular structures, but the relation is murky at best.

The Reviewer raises the reasonable question about how suitable it is to characterise the mechanics of gastrulation using different moduli at different frequencies. In principle both are related by:

$$M = K + 4/3G,$$

where M is the longitudinal modulus, K the bulk modulus and G the shear modulus.

Year	Material	Outcome of modalities comparison	Compared Methods	Authors	DOI
2011	porcine and bovine eye lenses	agreement (correlation >0.9)	BM (M) vs. AFM (E)	Scarcelli G. et al.	https://doi.org/10.1016/j.bpj.2011.08.008
2012	bovine cornea	agreement	BM (M) vs. IOP (E)	Scarcelli G. et al.	https://doi.org/10.1167/iovs.11-8281
2015	mouse fibroblast cells (NIH 3T3)	agreement (correlation >0.9)	BM (M) vs. AFM (E)	Scarcelli G. et al.	https://doi.org/10.1038/nmeth.3616
2017	bovine retina	agreement (correlation >0.9)	BM (M) vs. AFM (E)	Weber S. et al.	https://doi.org/10.1088/1478-3975/aa6d18
2020	Gelatin phantoms and ex vivo porcine lenses	agreement	BM (M) vs. OCE (E)	Ambekar YS et al.	https://doi.org/10.1364/BOE.387361
2022	human glioblastoma cells (U87)	agreement	BM (M) vs. optical tweezers (G)	Nikolić M. et al.	https://doi.org/10.1016/j.bpj.2022.09.002
2023	polyacrylamide hydrogels	agreement	BM (M) vs. rheometer (G)	Rodriguez López R. et al.	https://doi.org/10.1021/acs.biomac.3c01073
2023	MCF10A and MCF10AT1k.cl2, assayed as single cells and spheroids	agreement	BM (M) vs. AFM (E)	Zhang J. et al.	https://doi.org/10.1038/s41592-023-01816-z
2024	rat retina	disagreement	BM (M) vs. AFM (E)	Gutmann M. et al.	https://doi.org/10.1088/2515-7647/ad5ae3

For an isotropic sample, such as water (which makes up most of the cell), the speed of sound and therefore, the longitudinal modulus, is constant across frequencies. In principle it can also be related to the Young's modulus through the Poisson's ratio. Yet we acknowledge that formal relations are difficult to use in anisotropic biological systems. This is because the Poisson's ratio is strongly frequency-dependent and it is challenging to measure at high frequencies.

However, the physical origin of the correlation between a high frequency longitudinal modulus and lower frequency mechanical measurements has been characterised and discussed for simpler systems such as hydrogels (Rodriquez Lopez R. Biomacromolecules 2023). For more complex biomaterials, as the Reviewer correctly points out, there are empirical correlations between M at high frequencies, using Brillouin microscopy, and E or G (Shear modulus) at low frequencies, using AFM, optical tweezers, rheometers or OCE (Optical Coherence Elastography), in different experimental conditions. Some studies have quantitatively compared these measurements, predominantly finding agreement between high- and low-frequency modalities across biological samples and tissue types, except for one study on the rat retina, which reports results that actually differ from findings in the eyes of other organisms. To aid the Reviewer, we have compiled a comprehensive table below. Despite these largely consistent correlations, we want to stress that we regard Brillouin microscopy and AFM as complementary tools to characterise the mechanical properties of biological systems, since the two techniques measure different components of the same elastic tensor, and at different frequencies. The table below summarises the studies in which these comparisons have been conducted and their outcomes:

We would like to stress that we did not state in our paper that our results rest on such empirical correlations. On the contrary, we believe that the measurements of the (Brillouin) longitudinal modulus provide new and valuable information on their own on the mechanical characterization of the highly dynamic process of gastrulation.

Action

We have introduced the following sentences throughout the manuscript to increase the clarity about the point raised by the Reviewer:

In the introduction:

(...)Second, Brillouin microscopy measures mechanical properties at GHz frequencies, probing material responses on timescales several orders of magnitude smaller (~nanoseconds) than those of cellular-level biological processes (~milliseconds). Despite this, direct comparisons between the two techniques show high empirical correlations between these two regimes (Scarcelli G. et al. 2011 Biophysical J.; Scarcelli G. et al. Nat Methods 2015; Weber I. et al Phys Biology 2017; Zhang J. et al. Nat Methods 2023). Typically a power law can be observed, both in cells (Scarcelli G. et al. Nat Methods 2015 and tissues (Scarcelli G. et al. 2011 Biophysical J.), with an exponent varying in the range of 0.02-0.09. That implies that the relative change in quasi static modulus is often 10 to 50 times larger than the measured relative change in the Brillouin modulus. (...)

In the Results:

(...) Previous studies have shown a link between actin dynamics and the Brillouin shift (Scarcelli G. et al. Nat Methods 2015; Zhang J. et al. Small 2020; Coker Z.N. Photonix 2024).(...)

Additionally, if the Reviewer finds it suitable, we would be pleased to add the table above as Supplementary Table 1. This table may aid readers, specialist or not in the mechanobiology field, in understanding the outcomes of the correlations between high and low frequency regimes.

11. The distant relation between the two classes of moduli is not articulated in the abstract, introduction or Results sections as far as I can see. It is dealt with in Discussion (2nd para), but this is rather late.

The above comments make us realize that we haven't properly introduced the background to Brillouin microscopy in an adequate manner for the general reader. We fully agree with the Reviewer that this information is very important.

Action

We have revised the manuscript to give an elaborate definition of the elasticity tensor in the Introduction:

(...)To characterise the material properties, the full elasticity tensor is required, which is inherently dependent on both the probing frequency as well as the wavevector (directionality). Even in an isotropic material, the elasticity tensor comprises multiple independent components, encompassing tensile, shear, and longitudinal moduli, each associated with different stress and strain directions.(...)

as well as point out the peculiarities of Brillouin microscopy where appropriate (we refer the reviewer to points 9 and 10). We hope this addresses the comments of the Reviewer.

12. Relative changes in BS reported here are very small, typically of order 0.5 % (e.g. from 5.17 GHz to 5.18 GHz, Fig. 2a). This presumably corresponds to ~ 0.5 % relative changes in high frequency moduli. Even if one accepts an empirical relationship with cell stiffnesses, it is unclear if such tiny ~ 0.5 % changes in cell stiffness could be biologically relevant.

As mentioned above, the longitudinal modulus can be empirically related to the quasi-static Young's modulus via a power law with an exponent varying in the range 0.02-0.09, as established through empirical data both in cells (Scarcelli G. et al. Nat Methods 2015)] and tissues (Scarcelli G. et al. Biophysical J. 2011). This implies that the relative change in Young's modulus is 10 to 50 times larger than the measured relative change in longitudinal modulus and hence BS (i.e. ~0.5% in longitudinal modulus can correspond up to ~25% in quasi-static modulus). Therefore we believe that even such relatively small changes of the high-frequency longitudinal modulus (on the percent level) can actually be biologically relevant.

Action

To highlight this for the general reader, we have incorporated this relevant relationship in the Introduction section as follows:

(...) Typically a power law can be observed, both in cells (Scarcelli G. et al. Nat Methods 2015) and tissues (Scarcelli G. et al. Biophysical J. 2011), with an exponent varying in the range of 0.02-0.09. That implies that the relative change in quasi static modulus is often 10 to 50 times larger than the measured relative change in the Brillouin modulus.(...)

13. It is argued that the change in BS is attributable to microtubules, based on colcemid-treated embryos that exhibit a lower BS than control (Fig. 4f). However, the furrow becomes more

open; it is possible the more open furrow directly alters the BS, rather than a direct effect from microtubules (i.e. any change that alters the furrow shape, microtubule-related or other, may conceivably affect the BS similarly).

The Reviewer rightly points out that the reduction in the Brillouin shift following microtubule depolymerisation could result either directly from the loss of a structural, mechanical function of the microtubules themselves, or indirectly, from the reported morphogenetic defects that arise in VFF upon microtubule depolymerisation (wider and shallow furrow and impaired mesoderm invagination), as shown in our and previous studies (Ko C et al. JCB 2018; Gomez JM et al. eLife 2024). While the alignment of microtubules along the apical-basal axis, coinciding with the Brillouin probing direction, is consistent with an increase in the Brillouin shift (Fig. 4c-e; see also point 4), we cannot fully disentangle whether the observed changes in the Brillouin shift arise from microtubule-dependent mechanical properties or from their role in facilitating mesoderm invagination.

As we had stated in the introduction, cell shape results both from the forces (internal and external) acting on a cell and the material properties of that cell. In the context a morphogenetic event, perturbations on cellular components that are directly engaged in the generation and transmission of forces, or, in defining the cell's material properties are expected to affect both the morphogenetic event itself as well as the a particular aspect of the cell's mechanical properties (forces/material properties). For example, perturbations on the dorso-ventral patterning signalling cascade, which controls Myosin light chain activation during VFF, affect both the mechanics of mesoderm cells (their contractility) as well as VFF (Leptin M. and Grunewald B., Development 1990; Fuse N. et al., Development 2013; Gomez JM et al., eLife 2024). Hence, the coupling between the mechanical inputs (forces and material properties) and the progression of the morphogenetic event itself is intrinsic to the study of cell shape transitions.

Action

We now discuss the possibility that microtubules directly or indirectly determine an increase in the Brillouin shift in the Discussion section:

(...)Instead, our results point to a mechanical role of microtubules during VFF, as Colcemid treatment impairs the Brillouin shift increase within central mesodermal cells. However, the effect of microtubules on the Brillouin shift during VFF could be manifold: First, they could have a direct effect, acting as structural mechanoeffectors that transmit forces (Ko C. et al. JCB 2018) or resist compression (Li, Y., Kučera, O., Cuvelier, D. et al. Nat Methods 2023; Ju, R.J., Falconer, A.D., Schmidt, C.J. et al. Nat Cell Biol 2024). Alternatively, their effect could be indirect, by perturbing a downstream process facilitated by microtubules. Microtubule depolymerisation impairs cell shape and nuclear position across dorso-ventral cell populations (Ko C. et al. JCB 2018; Gomez JM et al. eLife 2024), and specifically, prevents mesoderm invagination. Therefore, it remains a possibility that perturbing mesoderm fold morphology and invagination in Colcemid-treated embryos (Fig. 4f) causes the reduced Brillouin shift, rather than the structural function of microtubules themselves.(...)

14. Throughout the mathematical model section the “optimal” furrow formation is referred to as a standard against which to compare model results. However no experimental evidence is cited for furrow depth or shape to justify the “optimal” definition

We realise this is an important point we had failed to properly explain in the first version of this manuscript. In *Drosophila melanogaster*, the formation of the ventral furrow is followed by an invagination event at gastrulation stage. Earlier work found that *Drosophila* gastrulation produces deeper folds in comparison to other species, such as *Chironomus riparius* (Urbansky S. et al., eLife 2016) and other studies associated shallow furrows with failed mesoderm invagination, such as in *twist* mutants (Supplementary Video 12; Leptin M. and Grunewald B., Development 1991), by blocking neuroectoderm -lateral cells- displacement towards the ventral midline using cauterisation (Rauzi M. et al, Nat Comms 2015) or in the case of Colcemid treatment (Ko C. et al. JCB 2018; Gomez JM et al., eLife 2024). Altogether, these indicate that, the deeper the furrow the more likely is invagination to occur, and thus, gastrulation to be completed. Hence, furrow depth becomes a natural proxy for successful gastrulation events driven by invagination.

We would like to highlight that for the physical model presented in Fig. 5 we focused on the initial step of gastrulation, the folding of the mesoderm. A model that would also incorporate the subsequent invagination step requires the incorporation of additional processes such as the neuroectoderm movement towards the ventral midline (see Rauzi M. et al. Nat Comms 2016; Supplementary Video 1) and the lateral constriction of the mesodermal cells (Gracia M. et al. Nat Comms 2019), two processes that happen after fold formation, and are out of the scope of this physical model.

Action

To make this point clearer, we have now explained the rationale for using the furrow depth as a proxy of the 'success' of the folding outcome, as well as a citation to the abovementioned paper in the Results section and within the Supplementary Note:

(...)A comparative analysis of invagination and ingression events revealed that mesoderm invagination during *Drosophila* VFF requires a deeper fold than mesoderm ingression in *Chironomus riparius*(Urbansky S. et al., eLife 2016). Moreover, both this study and previous work (Leptin M. & Grunewald B., Development 1991; Ko C. et al., JCB 2018; Gómez J.M. et al., eLife 2024) showed that smaller or shallow mesoderm folds correlate with failed invagination. Consequently, we used fold depth - here furrow depth- (Fig. 5a) as a metric to quantify the extent of fold formation and, therefore, as a proxy for the chance of successful invagination.(...)

Minor issues:

15. Fig. 1h shows linear regression results for the Brillouin shifts of the dorsal ectoderm and neuroectoderm, to demonstrate they have the same slope. However, to demonstrate this needs two distinct regression fits, with a statistical test to compare the two fitted slopes.

We kindly refer the Reviewer to the point 1 of this letter.

The Reviewer is conceptually correct, and indeed we had tested statistically whether one linear regression line can fit both the neuroectoderm and ectoderm in the first submitted version of this manuscript. These analyses were conducted using Prism GraphPad, which compares the *slopes* and the *y-intercepts* obtained for the individual linear regressions of the neuroectoderm and

ectoderm, with a combined/general linear regression model for both cell types to determine if the *slope* and the *y-intercept* of the combined model are statistically different from those of the individual regressions (i.e. neuroectoderm/ectoderm). Thus, the p values we had reported in the results section correspond to this statistical test, indicating that neither the *slope* nor the *y-intersect* of the combined regression model differ significantly from the *slopes* and the *y-intersects* of the individual neuroectoderm and ectoderm regression lines. The methodology implemented in Prism GraphPad is the one described by Jerrold H. Zar in Biostatistical Analysis, Chapter 18, 2nd edition, Prentice-Hall (1984), which is conceptually-speaking, an Analysis of Covariance.

Action

We have now added new panels in **Supplementary Fig. 2a,b** to show the individual regressions for both ectodermal cell populations. Additionally, in the Results section, we added the following sentences with the clarification about the statistical comparison result:

(...) Linear regression analyses of neuroectoderm and dorsal ectoderm cells suggested that the Brillouin shift decreases in a comparable manner in both populations during gastrulation (Supplementary Fig. 2a,b), with the dorsal ectoderm undergoing further reduction in the Brillouin shift -compared to the neuroectoderm- until completion of squamous morphogenesis (Fig. 1e,g). To study whether their Brillouin shift dynamics were truly comparable, we tested if the individual regressions for the neuroectoderm and ectoderm could be described by a combined linear regression model. This showed that the evolution over time of the Brillouin shift in both cell types can be fit by the same line (Brillouin Shift (MHz) = $-0.33 \times t + 0.28$; slope comparison: $p=0.084$; intercept comparison: $p=0.795$). These results support a model in which cells along the dorsal-ventral axis exhibit two types of material properties: biphasic Brillouin shift behavior in the mesoderm and softening at similar rates across ectodermal cells.(...)

In the Methods section, we added the following sentences with the clarification about the combined regression model and the statistical comparison between the linear regression of the neuroectoderm and ectoderm:

(...)In Fig. 1h we conducted a combined linear regression model (Prism Graphpad V10.2.3) to test if the Brillouin shift dynamics in neuroectoderm and dorsal ectoderm cells can be fit with the same linear regression line. For this, we performed the combined linear regression analysis on the time course of the Brillouin shift measurements within the neuroectoderm and the dorsal ectoderm cells for each embryo between timepoints -0:04:16 hour (end of cellularisation) and 01:01:23 hour (ongoing squamous morphogenesis). Prism Graphpad V10.2.3 evaluates if the combined regression model yields an *interaction coefficient* between the neuroectoderm and dorsal ectoderm cells that is equal to 0 (zero= slope and/or intercept same for both cell types) (ref), both for the slope and the y-intercept.(...)

16. Fig. 2g shows Pearson correlations between Brillouin shift and the lengths in the apical-basal and stretching axes, respectively, from Fig. 2f. Since in Fig. 2f the curves have strong trends in common, I believe that measuring the Pearson correlation is not appropriate for this situation.

The Reviewer was right about the suitability of conducting correlation analyses without proper detrending.

Action

We have carefully reassessed the impact of time-dependent trends on the correlation between shape parameters and the Brillouin shift.

To proceed with the detrending, we first extracted the measurements of cell shape and Brillouin shift of averaged cells (Fig. 2f) for the time windows where there is detectable change: 23:17'-59:16' for the apical-basal axis and 29:38'-57:09' for the stretching axis. Stationarity of the time series was achieved by differencing the series once and was confirmed using the KPSS (Kwiatkowski–Phillips–Schmidt–Shin) test. An ARMA (autoregressive–moving-average) model was then fitted to each series using the `auto.arima` function from the R package `forecast`. The choices of selected parameters for the models were consistent with visual inspection of the autocorrelation and partial autocorrelation plots. Spearman correlations were then computed between model residuals.

Spearman correlations after detrending:

	apical-basal axis	stretching axis
time-window 2 (23:17'-59:16')	0.43	nd
time-window 3 (29:38'-57:09')	nd	-0.36

The R code for this analysis is available at:

https://git.embl.de/heriche/brillouin_shift_time_series

We have added the above mentioned link for the R code in the Methods section, subsection Image Quantification and updated the methodology accordingly explaining the detrending methodology as follows:

(...)To analyse the correlation between the cell shape parameters (apical-basal and stretching axes length) and the Brillouin shift during the time course of squamous morphogenesis we detrended the data from the averaged cells (Fig. 2f). To proceed with the detrending, we first extracted the measurements for the time windows where there is detectable change: 23:17'-59:16' for the apical-basal cell axis length and 29:38'-57:09' for the stretching axis length. Stationarity of the time series was achieved by differencing the series once and was confirmed using the KPSS (Kwiatkowski–Phillips–Schmidt–Shin) test. An ARMA (autoregressive – moving - average) model was then fitted to each series using the `'auto.arima'` function from the R package `forecast`. The choices of selected parameters for the models were consistent with visual inspection of the autocorrelation and partial autocorrelation plots. Spearman correlations were then computed between model residuals. The R code for this analysis is available at: https://git.embl.de/heriche/brillouin_shift_time_series (...)

Additionally, we have rephrased the corresponding paragraph in the Results section as follows:

(...)We detected changes in the Brillouin shift shortly after alterations in the apical-basal axis length (Fig. 2f, t = 23:17 min), which persisted as the stretching axis lengthened (Fig. 2f, from t = 29:38 min onwards). The Brillouin shift plateaued (Fig. 2f, t = 59:16 min) around the time when both the apical-basal (Fig. 2f, t = 59:16 min) and stretching axes (Fig. 2f, t = 57:09 min) ceased changing. After detrending the time evolution of the Brillouin shift and cell shape parameters (see Methods), we found a correlation of 0.43 between the apical-basal axis and the Brillouin shift, and an anticorrelation of -0.36 between the stretching axis and the Brillouin shift. Overall, our results show that cell shape changes not only coincide with Brillouin shift variations, but also retain a time-independent association, which is slightly stronger for the apical-basal axis.(...)

In sum, these findings confirm that time-related trends previously influenced the observed correlations. After detrending, the association between shape descriptors and the Brillouin shift remains evident, with a stronger link emerging between reductions in apical-basal length and decreases in the Brillouin shift.

17. Last paragraph of “The role of microtubules in the determination of mechanical properties during VFF.” Fig. 4e is referred to for results on twist mutant embryos, but as far as I can see Fig. 4e does not involve twist mutants.

The reference to the correct Figure has been replaced in the revised manuscript.

Reviewer #3 (Remarks to the Author):

Gomez et al. describe the experimental characterization of embryonic cells during *Drosophila* gastrulation using line-scan Brillouin microscopy. The Brillouin frequency shifts exhibited spatial and temporal variations on a time scale of minutes during morphogenesis. These changes were compared with the distribution and concentration of several structural molecules, such as actin fibers and microtubules. While the Brillouin data presented are technically sound, novel, and interesting, the interpretation of these images in terms of cell stiffness, and their connection to structural molecules, requires more careful consideration and supporting data.

We thank the Reviewer for their positive assessment of the novelty and interest of the underlying Brillouin data. However, we also realize that a more elaborate and detailed introduction about the methodology and the nature of the longitudinal moduli, and discussion of the Brillouin data is warranted in order to avoid misinterpretations. Below we provide more details on the individual points raised by the Reviewer.

18. As the authors mention, Brillouin microscopy measures a specific mechanical property, namely the longitudinal modulus (both real and imaginary components). Throughout the manuscript, the Brillouin frequency shifts are often referred to as “stiffness” or “mechanical properties,” terms also used in the context of AFM and other mechanical measurements from the literature, which typically refer to “shear modulus” or “Young’s modulus.” However, the longitudinal modulus and shear modulus are distinct properties. Although they can be correlated in biological tissues, this relationship is phenomenological rather than fundamental. This distinction is insufficiently clarified in the manuscript. For example, in the very first sentence of the abstract, the mechanical properties that affect cell morphology in response to cellular forces mostly pertain to shear moduli. The role of longitudinal moduli in determining cell shape is thought to be minimal, if not negligible.

We kindly refer the Reviewer to the general response and to points 9 and 10.

We fully agree with the Reviewer's statement and acknowledge that the distinction between the longitudinal modulus measured by Brillouin microscopy and other moduli as measured by AFM etc. have been insufficiently clarified. We note that while the longitudinal and shear modulus can in principle be related to one another (see Prevedel R. Nat Methods 2019, Box 4, as well as our response to point 10 above), doing so at the same frequency scale is difficult, since BM measures at GHz and shear moduli are typically assessed at much lower frequencies. In this sense the Reviewer is correct that the relationship is mostly phenomenological. We also agree that most of the work conducted in the field of mechanobiology so far has measured either the Young's or shear moduli and that they are fundamentally distinct, even though they can all be derived from the elastic tensor (which notably is the quantity that one would need to measure to have a fully mechanical characterisation of an arbitrary material). We have revised the manuscript accordingly and now clearly spell out the distinction, as suggested by the Reviewer.

We would also like to address the Reviewer's comment regarding the role of longitudinal moduli in determining cellular morphology. While the Reviewer suggests that this relationship may be minimal or negligible, in all dorso-ventral cell populations we observed changes in shape along the longitudinal direction (the apical-basal cell axis) albeit strongly in mesodermal and ectodermal cells. Therefore, the mechanical characterisation using the longitudinal modulus is -in principle- relevant for the morphogenetic events that constitute *Drosophila* gastrulation. Furthermore, several studies have shown that Brillouin shifts are sufficiently sensitive to detect changes in the organisation and density of cytoskeletal components, such as F-actin, both at the cellular and tissue levels (Scarcelli G. et al. Biophys J 2011 and Nat Methods 2015). Recent work by Zhang J. et al. (2020) investigated nuclear mechanics in intact cells and identified a correlation between changes in the nuclear longitudinal modulus, nuclear shape, and perturbations of cellular components, including the cellular cytoskeleton and the nuclear lamina (lamin A/C). Given the well-established role of the cytoskeleton in regulating cell shape, both for actin (Salbreux G., Charras G. and Paluch E. Trends in Cell Biology 2012, Perez-Vale KZ, Peifer M. Development 2020, Kelkar M, Bohec P, Charras G. Curr Opin Cell Biol. 2020) and microtubules (Röper K. Philos Trans R Soc Lond B Biol Sci 2020, Liu T. et al. JCS 2010), it follows that the Brillouin shift and longitudinal modulus are indeed relevant measures for characterising the mechanics underlying cell shape transitions.

Action

In the revised manuscript we now included a broader overview and discussion about the different moduli and how they can (not) be related to each other, as well as an improved background description of the cellular processes that have been characterised using the longitudinal modulus.

In the Introduction we have now clearly explained the main differences between Brillouin microscopy and more established methods (as AFM):

(...)The shift in energy of the Brillouin-scattered light is related to the sound velocity of the probed material, which is dependent on its material properties (Prevedel R. et al. Nat Methods 2019; Bouvet P. et al. BiorXiv 2024). To characterise the material properties, the full elasticity tensor is required, which is inherently dependent on both the probing frequency as well as the wavevector (directionality). Even in an isotropic material, the elasticity tensor comprises multiple independent components, encompassing tensile, shear, and longitudinal moduli, each associated with different stress and strain directions. The longitudinal modulus, defined as the ratio between the uniaxial stress

to strain, can be calculated from the Brillouin shift if the refractive index and mass density are known. However, even in the absence of these parameters, the Brillouin spectrum can be used as a proxy of visco-elastic properties (Schüßler R. et al. Biophysical J 2018). Indeed in this study we report the Brillouin shift as a proxy to the longitudinal modulus, as commonly done in the field.(...)

and:

(...)Brillouin microscopy differs from more established rheological methods in two main aspects. First, Brillouin microscopy measures the longitudinal modulus whereas established rheological methods such as AFM measure the Young's and/or Shear Moduli(Prevedel R. et al Nat Methods 2019; Petridou N. et al. EMBO J 2019). Because the longitudinal modulus is typically not accessible by standard techniques (e.g. AFM), it has been less characterised in biological systems. Notably, the Young's, shear and longitudinal moduli describe different responses of materials to particular stresses. They can all be derived from the elasticity tensor (Bouvet P. et al. BiorXiv 2024) but measuring all the independent components is practically not feasible in anisotropic materials such as many biological systems. Second, Brillouin microscopy measures mechanical properties at GHz frequencies, probing material responses on timescales several orders of magnitude smaller (~nanoseconds) than those of cellular-level biological processes (~milliseconds). Despite this, direct comparisons between the two techniques show high empirical correlations between these two regimes(Scarcelli G. et al. 2011 Biophysical J.; Scarcelli G. et al. Nat Methods 2015; Weber I. et al Phys Biology 2017; Zhang J. et al. Nat Methods 2023).(...)

and:

(...)While its limitations and assumptions must be carefully considered, Brillouin microscopy provides a powerful platform for exploring the mechanobiology of developmental processes with subcellular resolution, providing insightful information about the material properties of (sub-) as well as cellular compartments. The utility of Brillouin microscopy is particularly evident in scenarios where the target tissue is inaccessible to other contact-based techniques, such as AFM. This is exemplified in studies of *Drosophila* gastrulating embryos, where the blastoderm is encased by the vitelline membrane, rendering direct mechanical access and measurements of the underlying blastoderm unfeasible.(...)

19. The authors appear to assume that higher Brillouin frequencies correspond to higher stiffness and attempt to interpret the data using established knowledge around shear stiffness, such as the relationship between cell deformability and structural molecules. However, since the correlation between longitudinal and shear moduli is not guaranteed and can be influenced by other physical properties—particularly the solid-to-fluid ratio—this approach risks leading to erroneous conclusions.

The relationship between the Brillouin shift and the longitudinal modulus, is indeed well established in the literature (e.g. Vaughan JM and Randall JT, Nature 1980, Palombo F. and Fioretto D. Chemical Reviews 2019, Prevedel R, et al., Nat Methods 2019). We also highlight that at no point in the paper we associate the Brillouin measurements with shear *stiffness* (i.e. shear modulus), and that we do not consider such shear moduli in our analysis or interpretation. In fact, we do not follow why the Reviewer uses the shear moduli as a reference or necessity for the description of any mechanical change in our system, in particular during morphogenesis. To our knowledge, there is no evidence in the literature that supports the *unique* role for shear moduli.

With respect to the solid-to-fluid ratio, the Reviewer is correct that they may influence the calculation of the longitudinal modulus through variations of the density and refractive index

parameters. We refer the Reviewer to point **21** below, where we provide a detailed answer to this point.

In conclusion, we would like to highlight again that our work investigates the dynamic changes of longitudinal modulus of cells involved in VFF - we believe this contributes valuable mechanical information to the description of this highly complex, mechanical process. To ensure that our work and data is not erroneously confused with other mechanical moduli, we have revised both the Introduction and Discussion sections.

20. Therefore, my primary concern is the accuracy of the physical model presented.

We assume the Reviewer refers to the physical model of ventral furrow formation (see Fig. 5). Even though this model is a toy-model, and it does not incorporate the absolute Brillouin shift values we have measured during the folding and invagination of the mesoderm, it does support our interpretation of the role of dynamic changes in longitudinal modulus profile in driving furrow formation. The results obtained using this model constitute an important addition to the manuscript, because the model confirms that differential contributions of sub-apical and sub-basal material properties, as well as dynamic changes in longitudinal moduli along the mesoderm, favours the formation of a deep furrow which is essential for successful invagination.

21. Even if a quantitative correlation between longitudinal and shear moduli were to exist in these embryos during morphogenesis, it could be phenomenological rather than mechanistic. For instance, intracellular protein concentration might increase alongside microtubule assembly, leading to decreased water content and a higher solid-to-fluid ratio, which would then increase the Brillouin frequency. In this case, shear stiffening by microtubules may not be the direct cause of the longitudinal modulus increase.

We kindly refer the Reviewer to the general response, and to points 9, 10 and 19.

We fully agree with the Reviewer that any correlation between the two moduli may be phenomenological rather than mechanistic. As previously mentioned in point **19**, it was not our goal to characterise the shear modulus along the dorsal-ventral cell populations at the time point of gastrulation.

The Reviewer is correct that changes in the solid-to-fluid ratio, e.g. through intracellular protein aggregation, could hypothetically explain the observed increase in Brillouin shift (by modifying the refractive index/density). However, we point out that such a localised and transient (with a max duration of 15 minutes) increase in protein abundance within the sub-apical compartment is highly unlikely, and has not yet been reported. Furthermore, the difference in protein abundance would need to be restricted not only to the sub-apical compartment but also to the central mesodermal cells, where we have observed most of the increase in the Brillouin shift during mesoderm folding and invagination (Fig 2a,b). However, at this point in development, the embryos are still undergoing maternal-to-zygotic transition (Arbeitman M., Iman F., Johnson E. et al. Science 2002; Harrison MM, Marsh AJ and Rushlow CA, Genetics 2023;). This means that with the exception of a small subset of the proteome, such as those governing dorsal-ventral patterning (Twist, Snail, Sog, Dpp; see Lynch D. and Roth S. Gen. Dev. 2011 and Gomez JM et al. eLife

2024) the majority of proteins are maternally provided (Gouw JW et al. MCP 2009). Furthermore, the *Drosophila* proteome at gastrulation stage shows no differences or very small differences in abundance across dorso-ventral cell populations, including many cytoskeletal components and its associated proteins (Gomez JM eLife 2024). Therefore, differences in protein abundance are expected to be minimal within each dorso-ventral cell population. Alternatively, the volume could change in central mesodermal cells. However, the volume of mesodermal cells was reported to be conserved during VFF (Gelbart M et al. PNAS 2012). Hence, we reason that the most likely source of the increase is *directly* or *indirectly* connected to microtubules.

Action

We acknowledge the important contribution of the Reviewer by raising this point, and have added a paragraph within the Discussion section about this point in the revised version of our manuscript:

(...)The entire (volumetric) mix of material content of a cell or tissue will determine its mechanical property; therefore a cell's solid-to-fluid ratio can have a pronounced effect that can be measured by Brillouin microscopy. For instance, the transient increase in the Brillouin shift within the sub-apical compartment of central mesodermal cells could, in principle, result from a local rise in protein abundance, altering the refractive index or density. However, such a transient (≤ 15 min) and localised increase in protein concentration within the sub-apical compartment is highly unlikely and has not been reported at the gastrulation stage. Moreover, this effect would need to be confined not only to the sub-apical compartment but also specifically to central mesodermal cells, where we observed the Brillouin shift increase during VFF (Fig. 2a,b). At this stage, the embryo is undergoing maternal-to-zygotic transition(ref), and the proteomes of dorsal-ventral cell populations display very small differences in abundance, including cytoskeletal components and its associated proteins (Gomez JM et al. eLife 2024). Thus, significant protein abundance differences are unlikely within the mesoderm. Additionally, central mesodermal cell volume remains constant during VFF (Gelbart M et al. PNAS 2012), reinforcing that the observed Brillouin shift dynamics likely reflect changes in cellular material properties.(...)

22. That said, the results clearly demonstrate that the longitudinal modulus of cells changes dynamically and spatiotemporally, which points to the potential utility of the technique for identifying such changes. In conclusion, while the technological advance enabling time-lapse imaging of whole organisms is impressive, the interpretation of the data in terms of cell deformation and structural molecules needs to be addressed more thoroughly and clearly.

Indeed, our aim was to characterise the changes in the longitudinal modulus during *Drosophila* gastrulation using Brillouin microscopy. Our mechanical results are restricted to the longitudinal modulus, and we hope the addition of a discussion paragraph to place the longitudinal modulus in context, as well as the clarification in the introduction and the more-detailed description of the relationship between the longitudinal modulus and the Brillouin shift will further balance our presentation as well as put our finding into broader context.